# Diverse but unique astrocytic phenotypes during embryonic stem cell differentiation, culturing and development

Kiara Freitag [1,2,7], Pascale Eede[1,6,7], Andranik Ivanov[3], Nele Sterczyk[1], Shirin Schneeberger[1,4,6], Tatiana Borodina[5], Sascha Sauer [5], Dieter Beule [3] & Frank L. Heppner [1,2,4 ✉]

Astrocytes are resident glial cells of the central nervous system (CNS) that play complex and heterogeneous roles in brain development, homeostasis and disease. Since their vast involvement in health and disease is becoming increasingly recognized, suitable and reliable tools for studying these cells in vivo and in vitro are of utmost importance. One of the key challenges hereby is to adequately mimic their context-dependent in vivo phenotypes and functions in vitro. To better understand the spectrum of astrocytic variations in defined settings we performed a side-by-side-comparison of murine embryonic stem cell (ESC)-derived astrocytes as well as primary neonatal and adult astrocytes, revealing major differences on a functional and transcriptomic level, specifically on proliferation, migration, calcium signaling and cilium activity. Our results highlight the need to carefully consider the choice of astrocyte origin and phenotype with respect to age, isolation and culture protocols based on the respective biological question.

[1] Charité – Universitätsmedizin Berlin, corporate member of Freie Universität Berlin and Humboldt-Universität zu Berlin, Department of Neuropathology, Charitéplatz 1, 10117 Berlin, Germany. [2] German Center for Neurodegenerative Diseases (DZNE) within the Helmholtz Association, Berlin, Germany. [3] Core Unit Bioinformatics, Berlin Institute of Health, Charité - University Hospital Berlin, 10117 Berlin, Germany. [4] Cluster of Excellence, NeuroCure, Berlin, Germany. [5] Scientific Genomics Platforms, Max Delbrück Center for Molecular Medicine (MDC) in the Helmholtz Society, Berlin, Germany and Berlin Institute of Health (BIH), Berlin, Germany. [6] Present address: Apollo Health Ventures, Schlüterstr. 36, 10629 Berlin, Germany. [7] These authors contributed equally: Kiara Freitag, Pascale Eede. ✉email: frank.heppner@charite.de

Astrocytes are central nervous system (CNS)-resident glial cells that have long been considered passive supporting cells, yet they have caught major attention due to their involvement in many neurological diseases. Astrocytes form vast intercellular networks within the brain that influence homeostatic cell metabolism and function[1,2]. As a main constituent of the neurovascular junction, astrocytes ensure the energy supply to the brain[3–5] whilst also controlling neuronal health by recycling neurotransmitters and by promoting the formation of neural networks[6]. Thus, due to their crucial role in brain homeostasis, dysfunction of astrocytes has a substantial impact on brain disorders, in particular upon neurodegenerative and neuroinflammatory diseases[7].

With an increasing requirement to properly assess astrocytes, the need for efficient isolation and culturing protocols keeping astrocytes close to their in vivo profile rises. Many methods to isolate and study murine astrocytes have been established, yet such approaches need to be carefully selected based on the underlying scientific question[8]. In vitro approaches using primary astrocytes provide a useful platform to dissect specific astrocytic functions and molecular mechanisms, however, maintaining an in vivo-like phenotype has been a major challenge. It is widely known that the classic astrocyte isolation technique using postnatal rodent brains based on McCarthy & DeVellis[9] induces a reactive astrocytic phenotype[10], thus changing the presumed in vivo profile of astrocytes. Similarly, astrocytic developmental stages, along with their functional and transcriptomic differences, are often not taken into account or compared between isolation techniques. Consequently, many studies assessing astrocyte functions only rely on cultures from neonatal astrocytes, thus neglecting the major functional changes that occur context-dependently during development and aging[11–13].

To generate a baseline reference of transcriptomic and functional changes of various astrocyte culture settings and origins, we compared magnetic-activated cell sorted (MACS) ACSA-2-positive murine astrocytes[14,15] isolated from (1) neonatal and from (2) adult wild-type mice to (3) astrocytes generated from mouse embryonic stem cells (ESC) (AGES)[16]. So far, thorough side-by-side comparisons of ESC- or induced pluripotent stem cell (iPSC)-derived astrocytes to their primary adult and neonatal counterparts taking phenotypic, functional, and transcriptomic characteristics into account, are to our knowledge lacking. Additionally, we investigated how cellular properties of neonatal and adult astrocytes change upon culturing in order to highlight which in vivo functions are specifically altered in an in vitro setting. Analysis of astrocyte markers, transcriptomic profiles, and functional properties revealed major differences between the various astrocyte populations. Whilst functions related to trophic support, such as synaptic vesicle transport and dendritic spine development, were lost upon culturing of primary astrocytes, key age-specific differences in extracellular matrix regulation were retained. AGES displayed distinctive transcriptomic and functional signatures resembling astrocytic characteristics that did not fully align with primary astrocyte profiles and most likely represent an intermediate state between primary cells and neural stem cells (NSCs). These data highlight the importance of carefully aligning experimental requirements to the underlying biological question when assessing astrocytic properties in vitro and ex vivo.

## Results and discussion
**Distinct astrocytic marker profiles between AGES, cultured, and directly isolated primary astrocytes.** Given their vital functions in health and disease, it is essential to have valid and robust astrocyte in vitro models allowing to assess their particular

contributions in physiological as well as CNS disease settings. Therefore, we compared the marker profiles of three widely used astrocytic cell types, namely of cultured and freshly isolated neonatal and adult astrocytes as well as AGES. Whole hemisphere neonatal astrocytes were isolated from 4- to 8-day-old C57BL/6 J wild-type mice by MACS using the neural tissue dissociation kit (NTDK) and anti-ACSA-2 magnetic microbeads (Fig. 1a). For adult astrocyte populations, cells were isolated from 100–140-day-old C57BL/6 J mice also using MACS, with the only change that the adult brain dissociation kit (ABDK) including debris and red blood cell removal steps was used (Fig. 1b). Cultured cells were used for downstream analyses after 7–10 days in vitro. For evaluating how efficiently AGES may replace primary astrocytes for the purpose of in vitro studies, murine ESCs (mESC) were differentiated into NSCs. After reaching purity (Supplementary Fig. 3a), NSCs were terminally differentiated into AGES within 3 to 5 days by adding bone morphogenetic protein 4 (BMP4)[16] (Fig. 1c). Fluorescence-activated cell sorting (FACS) for the astrocyte-specific cell surface marker ACSA-2 indicated that directly isolated astrocytes showed the highest level of ACSA-2-positive cells (97–98.7%), which was slightly reduced upon culturing and in AGES (Supplementary Figs. 1a, b, 2a–e). As ACSA-2 is also expressed by a few non-astrocytic cell types, such as glial progenitor cells, neural stem cells, and radial glia[15,17], we extended our characterization by assessing a wide range of established astrocyte markers.

Western blot and RT-qPCR revealed that directly isolated adult astrocytes expressed higher levels of the astrocytic calcium binding protein S100B (Fig. 1d and Supplementary Fig. 1d), while a stable protein expression of GLT-1 was found across cell types (Fig. 1e). Gene expression analysis of *Aldh1l1* and *Slc1a3* (*Glast*) revealed further differences between cultured and directly isolated adult and neonatal astrocytes as well as AGES (Supplementary Fig. 1e, f). Despite showing fewer ACSA-2-positive cells, fluorescent staining revealed that AGES have a 98% purity of GFAP-positive cells, showing a high differentiation efficiency. Cultured neonatal and adult astrocytes contained around 79% and 74% GFAP-positive cells (Supplementary Fig. 1c), yet GFAP-negative astrocyte subsets exist in both the neonatal and adult brain[18]. Western blot (Fig. 1f) and RT-qPCR analysis (Supplementary Fig. 1g) further confirmed differences in GFAP expression between AGES and primary astrocytes. All in all, the astrocyte marker profile of neonatal and adult astrocyte cultures mimic previously described maturation-dependent gene expression patterns such as an increase in *S100b* and a decline in *Gfap*, *Glast*, and *Aldh1l1* expression[11,19,20], whilst AGES do not fully reflect the marker profile of primary neonatal or adult astrocytes.

We next determined the level of contaminating microglia, neurons, and oligodendrocytes in our cultures. Microglia and neurons were not found within our cell isolations (Supplementary Fig. 1h–j and Supplementary Fig. 2a–e), whilst, in line with recent studies[14,21], 1–5% O4-stained oligodendrocytes were found in the primary astrocyte cultures with no O4-positive oligodendrocyte contaminations in the AGES culture (Supplementary Fig. 1l). Accordingly, directly isolated adult astrocytes showed the largest *Mbp* expression by RT-qPCR (Supplementary Fig. 1k), indicating that adult astrocyte isolations contained a slightly higher, however still low, oligodendrocyte contamination.

Certain astrocyte culturing protocols lead to the induction of (re) active astrocytes, which does not reflect the physiological state found in vivo. Thus, we analyzed gene expression levels of the known astrocyte (re)activity markers *C3*, *Serpina3n*, and *Mx1*. Compared to neonatal astrocytes cultures based on the McCarthy and de Vellis method (Vellis astrocytes)[9], known to induce reactive astrogliosis[22], expression of markers indicating (re)activity was low in our astrocyte cultures with the exception of a significantly higher

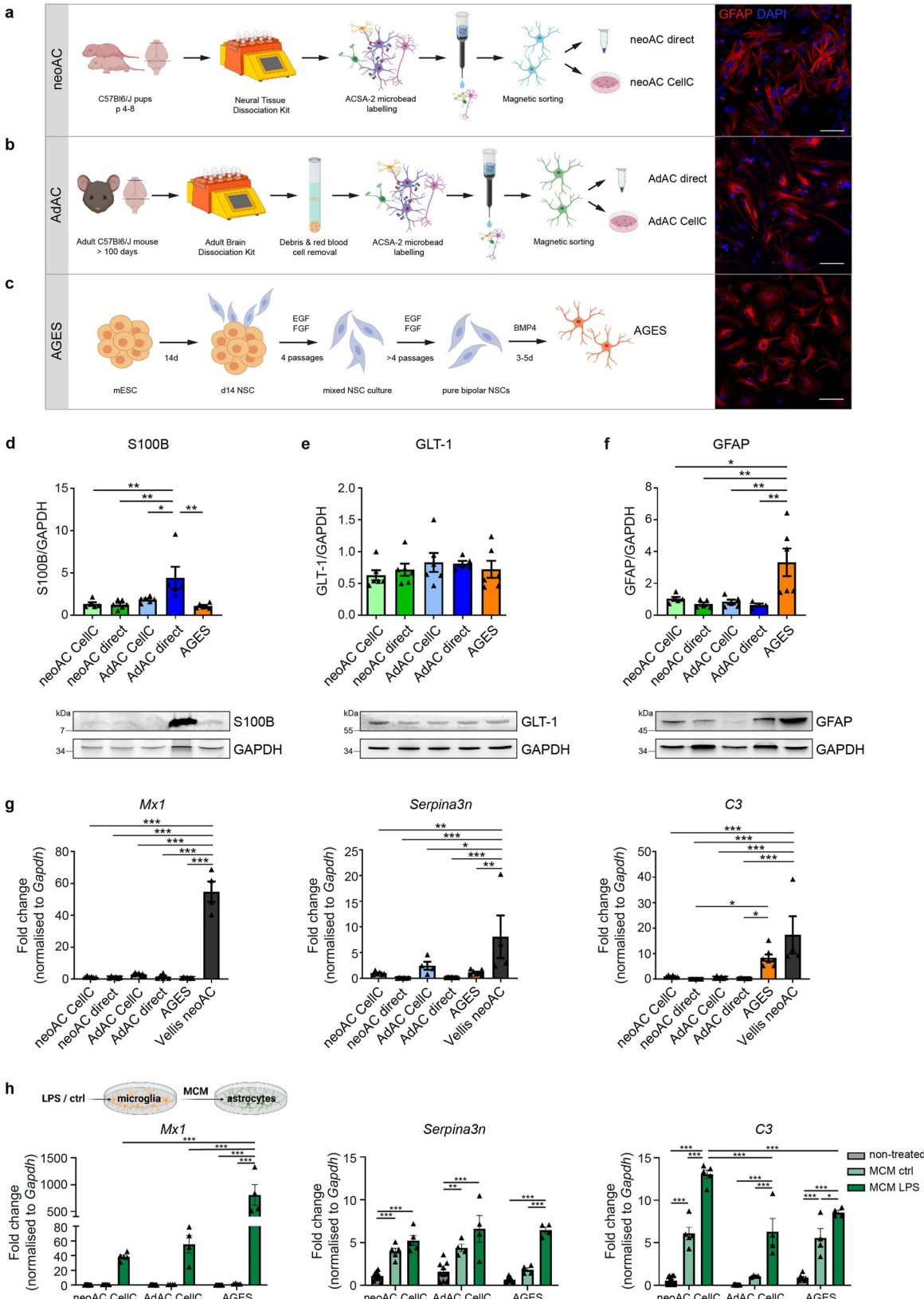

expression of *C3* in AGES (Fig. 1g), which is most likely a remnant of the C3-driven differentiation process[23]. The low (re)activity state of all astrocyte cultures became even more apparent by treating astrocytes with microglia-conditioned medium (MCM) from LPS- or non-treated microglia, which significantly upregulated the expression of *C3*, *Serpina3n*, and *Mx1*. The level of (re)activity was

highly different between all astrocyte cultures showing a 20-fold higher expression of *Mx1* in AGES and the highest upregulation of *C3* in neonatal astrocytes (Fig. 1h).

Taken together, our results highlight, when characterizing astrocyte cultures, a wide range of astrocytic and non-astrocytic markers should be used since the astrocyte subsets and astrocyte

**Fig. 1 Distinct astrocyte marker profiles between AGES, cultured, and directly isolated primary astrocytes. a** Neonatal astrocytes (neoAC) were isolated from 4- to 8-day-old C57BL/6 J mice by magnetic-activated cell sorting (MACS) using the neural tissue dissociation kit (NTDK) and anti-ACSA-2 magnetic microbeads. Cells were either snap-frozen (neoAC direct) or cultured (neoAC CellC). **b** Adult astrocytes (AdAC) were isolated from 100–140-day-old C57BL/6 J mice by MACS using the adult brain dissociation kit (ABDK), including a debris removal and red blood cell removal step followed by ACSA-2 microbead labeling. Cells were either snap-frozen (AdAC direct) or cultured (AdAC CellC). **c** Astrocytes generated from embryonic stem cells (AGES) were differentiated from mouse embryonic stem cells (mESCs) by creating neural stem cells (NSCs). For receiving pure bipolar NSC cultures, NSCs were cultured for at least eight passages in a medium containing the growth factors EGF and FGF. Differentiation into AGES was induced by adding bone morphogenetic protein 4 (BMP4) for 3 to 5 days. Fluorescent immunocytochemistry was performed for glial fibrillary acidic protein (GFAP). Images were taken by confocal microscopy. Scale bar = 50 μm. **d–f** Western blot analysis of all cell types was performed for the astrocyte marker proteins S100B (**d**), GLT-1 (**e**), and GFAP (**f**). Values were normalized to GAPDH and neoAC CellC. **g** Gene expression of the astrocyte (re)activity markers *Mx1* (***$P < 0.001$), *Serpina3n* (*$P = 0.0102$; **$P = 0.0024$; ***$P < 0.001$), and *C3* (AGES vs. neoAC direct *$P = 0.0470$; AGES vs. AdAC direct *$P = 0.0452$; ***$P < 0.001$) was determined by quantitative real-time PCR. neoAC CellC ($n = 5$), neoAC direct ($n = 6$), AdAC CellC ($n = 7$), AdAC direct ($n = 6$), AGES ($n = 7$), Vellis neoAC ($n = 4$). **h** Microglia were treated with LPS (1 μg/ml) for 24 h or remained non-treated. The microglia-conditioned medium with LPS (MCM LPS) or without LPS (MCM ctrl) was added to the respective astrocyte cultures for 24 h. Gene expression of *Mx1* (***$P < 0.001$), *Serpina3n* (**$P = 0.0038$; ***$P < 0.001$) and *C3* (*$P = 0.0204$; ***$P < 0.001$) was determined by quantitative real-time PCR. All expression values were normalized to the internal control *Gapdh* and the fold change compared to neoAC CellC was displayed; neoAC CellC non-treated ($n = 11$), neoAC MCM ctrl/ LPS ($n = 5$), AdAC CellC non-treated ($n = 10$), AdAC CellC MCM ctrl/ LPS ($n = 4$), AGES non-treated ($n = 7$), AGES MCM ctrl/LPS ($n = 4$). Mean ± SEM, ANOVA with Tukey's post hoc test.

marker profiles highly vary between differentiated AGES and primary astrocytes. Whilst the baseline level of (re)activity of our described cultures is low, the differences in the sensitivity to MCM-induced (re)activity in all cultured models should be considered when using these model systems.

**Similarities in glucose uptake, lactate release, and synaptosome uptake oppose differences in proliferation, migration, and calcium signaling in AGES and primary astrocyte cultures**. To investigate whether the differences seen in the astrocyte marker profile also implicate differences in functional properties, we assessed the maintenance of homeostatic functions of astrocytes in culture. Proliferation in astrocytes is known to cease with ageing[24], which we confirmed in a 5-ethynyl-2'-deoxyuridine (EdU)-based assay. Cultured adult astrocytes and AGES, as terminally-differentiated cells, showed no proliferation, while neonatal astrocytes exhibited a significantly higher proliferation rate (Fig. 2a), which was not influenced by the amount of dying cells (Fig. 2b). Compared to highly proliferating HEK293 cells and no-EdU controls, astrocyte proliferation was negligible (Supplementary Fig. 3b). To mimic wound healing upon tissue injury, a major feature of astrocytes in vivo, a confluent astrocyte layer was disrupted by creating a wound gap in vitro. Adult astrocytes migrated or sensed wound gaps significantly faster than neonatal astrocytes and AGES, with AGES exerting a very slow response rate (Fig. 2c–f and Supplementary Fig. 3c). Assessing further physiological functions of astrocytes such as metabolism of glucose and lactate[25] as well as synapse elimination[26] revealed no differences between all primary cultured astrocytes and AGES (Fig. 2g–i), emphasizing that AGES are a suitable in vitro model for assessing these physiological functions of astrocytes. To investigate calcium signaling as the main communication system of astrocytes[27], we performed live imaging of astrocytes incubated with the calcium indicator Fluo-4, which is coupled with an acetoxymethyl (AM) to ensure fluorescence only after cell entry (Fig. 2j, k). We found that AGES responded fastest to ATP stimulation (Fig. 2l), and their maximum fluorescent intensity i.e., the amount of calcium released, was highest compared to neonatal and adult astrocytes (Fig. 2m), which is in line with previous reports showing that the amplitude of the spontaneous calcium spike was significantly higher in human iPSC-astrocytes compared to primary astrocytes[28]. No differences in the duration of the calcium response were seen between the cell types (Fig. 2n). We also confirmed that the calcium response profile in AGES is different from that of NSCs (Supplementary Fig. 3d). Taken together, key differences in functions of AGES compared to

primary astrocyte cultures were found in proliferation, response to wound gaps and calcium release properties. Notably, it was suggested that even in instances of lesions, active cell migration might rather be a non-physiological function of astrocytes in vivo and may represent a behavior only acquired in culture systems in vitro. In vivo live imaging revealed astrocytes to either extend their processes toward the lesion or to selectively proliferate[29]. Despite showing differences to primary astrocytes, AGES might therefore be more suitable in modeling the in vivo-like migration behavior of astrocytes. Thus, it is of utmost relevance to consider the identified functional differences when choosing a model system for a respective biological investigation.

**Transcriptomic profiling of astrocytes reveals major differences introduced by cell culturing, age, and cell origin**. To obtain a deeper understanding of the similarities and differences of directly isolated and cultured neonatal and adult astrocytes as well as AGES and NSCs, we performed unbiased transcriptional profiling using RNA sequencing (RNA-seq). First, we aimed to delineate the differences we identified with regard to astrocyte marker expression as well as glial cell migration, calcium signaling, and synapse pruning on the transcriptomic level. We therefore performed principal component analysis (PCA) on functionally grouped genes and confirmed key differences in astrocyte marker gene expression between cell types and culture conditions (Supplementary Fig. 4a and Supplementary Data 1), supplementing our analysis of astrocyte markers on the mRNA and protein levels (Fig. 1d–f and Supplementary Fig. 1d–g). Additionally, we observed that gene signatures corresponding to glial cell migration (Supplementary Fig. 4b) and calcium signaling (Supplementary Fig. 4c) separated AGES from neonatal and adult primary astrocytes, matching the observed functional differences in wound gap and calcium responses. PCA analysis using genes related to synapse pruning, revealed clustering of AGES and all cultured astrocytes separately from NSCs, validating their similarities in the synapse uptake assay (Supplementary Fig. 4d). This analysis of functionally annotated gene clusters also indicated differences between cultured and directly isolated astrocytes supporting the notion that the migratory and synapse pruning capacity of astrocytes change upon culturing.

To obtain an unbiased view of the transcriptional differences between our various astrocyte populations, we performed a PCA using the complete transcriptomic signature of each cell type, which showed appreciable variation between cell types up to the fourth component (Fig. 3a–c). To functionally annotate the underlying genetic differences driving the PCA, we applied gene

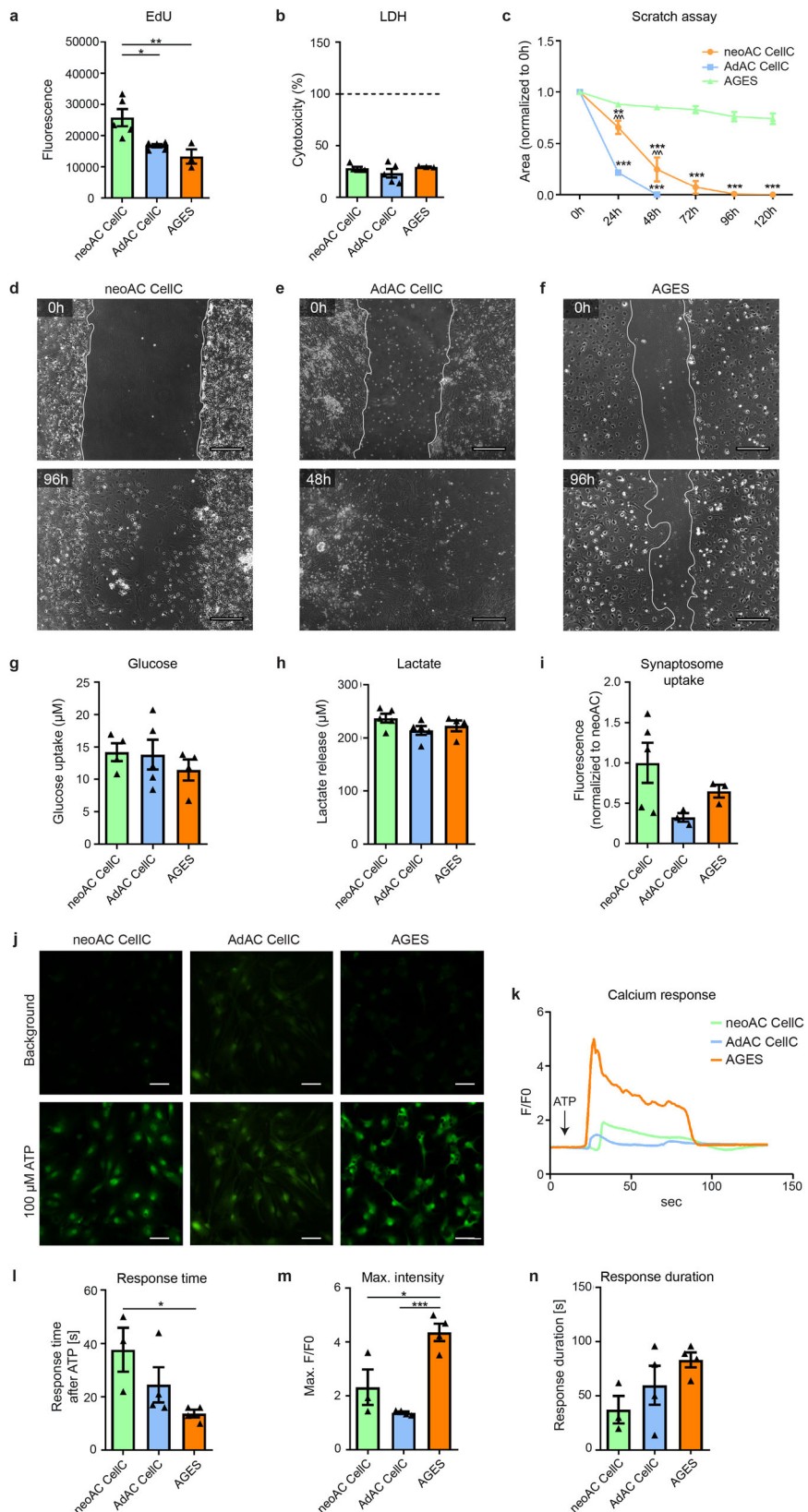

set enrichment analysis to the genes constituting the four PCs (Fig. 3d, e, Supplementary Fig. 4e, and Supplementary Data 2). Additionally, differentially expressed genes between two sets of cell types were identified and functionally annotated using the R tmod package for gene set enrichment analysis[30]. For each pairwise comparison and gene set, the fraction of upregulated genes is highlighted in red, the fraction of downregulated genes in blue and the non-regulated genes in gray. The intensity of each bar represents the $p$ value allowing for a side-by-side visualization of all comparisons (Fig. 3f, and Supplementary Data 3).

Many studies neglect the comparison of stem cell-derived models to either neonatal or adult primary astrocytes and only

**Fig. 2 Similarities in glucose uptake, lactate release, and synaptosome uptake oppose differences in proliferative, migratory, and calcium signaling of AGES and isolated astrocytes. a** Proliferation was measured by fluorescent immunolabeling for the thymidine analog EdU, which is incorporated during cell division. The fluorescent signal was normalized to background absorbance. ANOVA with Tukey´s post hoc test *$P = 0.0365$; **$P = 0.0068$.
**b** Cytotoxicity was measured by determining the levels of lactate dehydrogenase (LDH) release to the cell supernatant. As a positive control, each cell type was lysed with Triton-X to induce maximal LDH release per cell (=100%, dashed line). Medium served as a negative control and all values were plotted as percentages of the maximal LDH release. Mean ± SEM; neoAC CellC ($n = 5$), AdAC CellC ($n = 5$), and AGES ($n = 3$). ANOVA with Tukey´s post hoc test $P > 0.05$. **c** A scratch assay was used to determine the migratory behavior of cultured neonatal astrocytes (neoAC CellC), cultured adult astrocytes (AdAC CellC), and AGES. The area of the wound gap normalized to time point 0 h is shown over time. * significance compared to AGES; ^ significance compared to AdAC CellC. Mean ± SEM; neoAC CellC ($n = 4$), AdAC CellC ($n = 4$), and AGES ($n = 4$); Two-way ANOVA with Bonferroni post hoc test; **$P = 0.004$; ***$P < 0.001$; ^^^$P < 0.001$. **d–f** Representative phase contrast images of the wound gap at the starting point (0 h) and after 48 h/96 h are shown of **d** cultured neonatal astrocytes (neoAC CellC), **e** cultured adult astrocytes (AdAC CellC), and **f** AGES. Scale bar = 200 μm. **g** Glucose uptake (in μM) by cultured neonatal astrocytes (neoAC CellC), cultured adult astrocytes (AdAC CellC), and AGES was measured with a luminescence-based assay. Mean ± SEM; neoAC CellC ($n = 4$), AdAC CellC ($n = 5$), and AGES ($n = 4$). ANOVA with Tukey´s post hoc test $P > 0.05$. **h** To determine the lactate released by each cell type (in μM), lactate was measured in the cell supernatant with a luminescence-based assay. Mean ± SEM; neoAC CellC ($n = 5$), AdAC CellC ($n = 5$), AGES ($n = 4$). ANOVA with Tukey´s post hoc test $P > 0.05$. **i** pH-sensitive fluorescently labeled synaptosomes isolated from C57Bl/6 J mice were cocultured with cultured neonatal astrocytes (neoAC CellC), cultured adult astrocytes (AdAC CellC), and AGES. The fluorescence was measured and represents the uptake of synaptosomes. The fluorescent signals were normalized to cultured neonatal astrocytes. Mean ± SEM; neoAC CellC ($n = 5$), AdAC CellC ($n = 3$), and AGES ($n = 3$). ANOVA with Tukey´s post hoc test $P > 0.05$. **j–n** Calcium signaling of all three cell types was determined by calcium imaging using Fluo-4, AM as a calcium indicator, and 100 μM ATP as a stimulus for calcium release. **j** Representative images of cultured neonatal astrocytes (neoAC CellC), cultured adult astrocytes (AdAC CellC), and AGES are shown before ATP stimulation (=background) and after ATP stimulation. Scale bar = 50 μm. **k** The fluorescence intensity normalized to background fluorescence (F/F0) is shown over time for each cell type. **l** The time until cells responded with a calcium peak was measured. ANOVA with Tukey´s post hoc test *$P = 0.0493$. **m** The maximum fluorescence intensity was compared between all cell types. ANOVA with Tukey´s post hoc test *$P = 0.012$; ***$P < 0.001$. **n** The time until cells returned to baseline levels was determined. Mean ± SEM; neoAC CellC ($n = 3$), AdAC CellC ($n = 4$), AGES ($n = 4$); ANOVA with Tukey´s post hoc test; $P > 0.05$.

show comparisons to ESCs and NSCs. The few studies that did compare iPSC-derived astrocytes to primary cells[28,31] did not perform global assessments of phenotypic, functional, and transcriptional differences. Our transcriptomic analysis showed that primary neonatal and adult astrocytes expressed genes involved in cilium and axoneme function (PC1 left), with NSCs and AGES being characterized by increases in translation initiation, RNA processing, and proteasome activity (PC1 right) (Fig. 3b, d). tmod analysis comparing AGES to cultured primary cells identified the downregulation of genes related to DNA replication, cerebellar granular layer development and microtubule bundle formation and alterations in astrocyte cell differentiation genes (Fig. 3f). The negative regulation of adherens junction organization, cilium function and extracellular matrix regulation (Fig. 3f) could explain the observed differences in closing wound gaps in AGES, as they are crucial for a rapid cell–cell contact remodeling during wounding[32–34]. Underlining our findings, PCA analysis using published transcriptomic data of other murine stem cell-derived astrocytes[35,36] revealed a clear separation from neonatal and adult primary astrocytes indicating major gene expression changes between primary and stem cell-derived cells (Supplementary Fig. 4f). Our results indicate that AGES have a distinct functional and genetic profile to primary astrocytes and are not yet fully differentiated into astrocytes which is usually assisted and driven by the brain's microenvironment. In order for stem cell-derived astrocytes to serve as a tool to study a broad range of astrocyte functions, we need to fully understand their similarities and differences to primary astrocytes and culture protocols need to be adapted to more closely match the profiles of primary cells.

Culturing of astrocytes induced upregulation of genes enriched in translation initiation, RNA processing, proteasome activity, chemotaxis and vesicle transport (PC2 top/PC3 right), and the downregulation of genes involved in autophagy, inositol lipid pathway, and synaptic vesicle transport (PC2 bottom) and cell division (PC3 left, Fig. 3be). tmod analysis specifically revealed downregulation of cilium functions and microtubule formation and upregulation of cell division-related genes in adult astrocytes upon culturing. Neonatal astrocytes, on the other hand, upregulated microtubule formation and cilium functions

in vitro (Fig. 3f). One of the very few studies comparing directly isolated and cultured astrocytes found downregulation of Wnt and Notch signaling upon culturing of neonatal astrocytes isolated by immunopanning[12]. Interestingly, primary cilia alterations identified in our study are known to modulate the transduction of Wnt signaling[37]. To our knowledge, alterations introduced by cell culturing of adult astrocytes were not assessed so far. In summary, the changes introduced in vitro thus reflect the change in microenvironment upon culturing since the metabolic support of other cells is no longer required, however, age-specific differences are retained. AGES compared to NSCs downregulated genes indicative of cell cycle regulation and upregulated autophagy-related genes (PC2, Fig. 3b, d, f), which are common features of NSC differentiation[38,39] and indicate successful reprogramming into a new lineage with a modest proliferation behavior typical for astrocytes[12].

Age-specific differences included upregulated protein trafficking in adult astrocytes (PC4 top) and increased respiration, cell–cell adhesion, synapse regulation, and stem cell division in neonatal astrocytes (PC4 bottom) (Fig. 3c, e). tmod analysis revealed that, in vitro, neonatal astrocytes compared to adult astrocytes show a pronounced upregulation of cilium functions and microtubule formation as well as alterations in extracellular matrix regulation and astrocyte differentiation genes, correlating well with the observed differences in wound gap closing. Directly isolated neonatal astrocytes showed alterations in genes linked to cerebellar granular layer development and extracellular matrix regulation along with a downregulation in microtubule formation and cilium movement in contrast to directly isolated adult astrocytes (Fig. 3f). Thus, both the neonatal and adult astrocyte cultures retain key physiological functional characteristics carried out during postnatal neural development and aging respectively[6]. All in all, the transcriptomic differences we observed were driven by cell origin (primary vs. NSC-derived cells), culturing of cells, and developmental stage of primary cells (Supplementary Fig. 5).

**Distinct primary cilia morphology and appearance in AGES compared to primary astrocytes.** We identified striking differences in cilia function in all cell types. Specifically comparing the regulation of 35 genes in the GO term cilium movement revealed significant

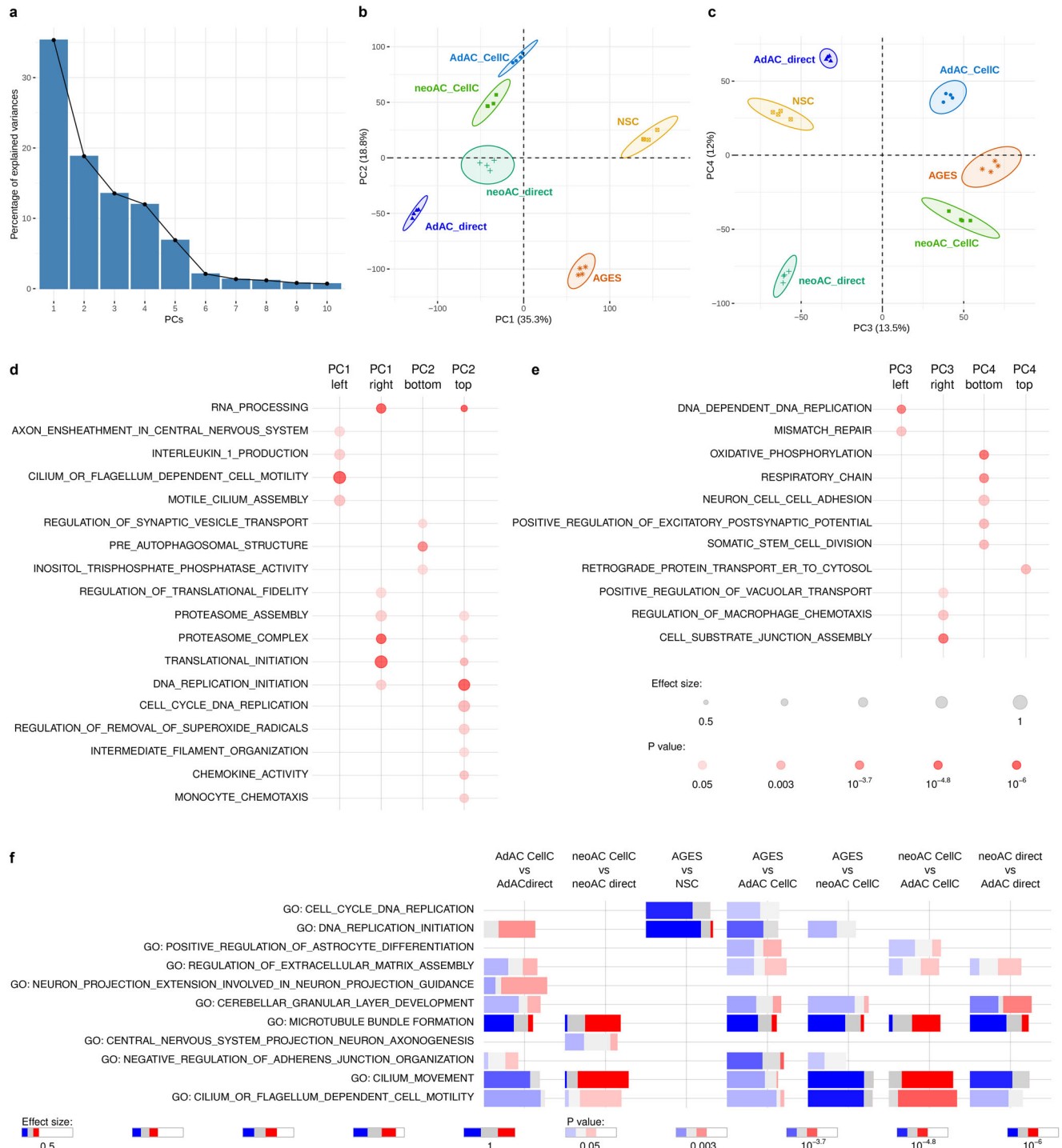

**Fig. 3 Transcriptomic profiling of astrocytes reveals major differences introduced by cell culturing, age, and cell origin.** RNA-seq was performed on cultured neonatal astrocytes (neoAC CellC; n = 4), directly isolated neonatal astrocytes (neoAC direct; n = 4), cultured adult astrocytes (AdAC CellC; n = 4), directly isolated adult astrocytes (AdAC direct; n = 4), AGES (n = 4), and NSCs (n = 4). **a–c** Principal component analysis (PCA) of RNA-seq results was performed and gene set enrichment analysis was applied to genes ordered by their PCA loadings. The variance explained by the components is shown on the x- and y-axis. **d, e** Selected gene ontology (GO) terms characterizing each principal component are shown. The size of each dot reflects the effect size, while the p value is visualized by color intensity. The effect size is the area under the curve (see Supplementary Fig. 4e). **f** Differential gene expression-based functional enrichment analysis between cell types was done using the R tmod package. Each bar presents the fraction of significantly up (red) and downregulated (blue) genes in that particular GO category.

differences between primary astrocytes and AGES with high similarities of gene regulation in AGES and NSC, indicating that AGES have not yet reached the cilia gene expression profile of primary astrocytes (Fig. 4a). Primary non-motile cilia have been described in both astrocytes[40] as well as human embryonic and neural stem cells[41,42] and are known to regulate cell division, neurogenesis, cell development, and the response to certain extracellular signals[40,43–45]. As differential stages of cilia development and altered ciliogenesis in

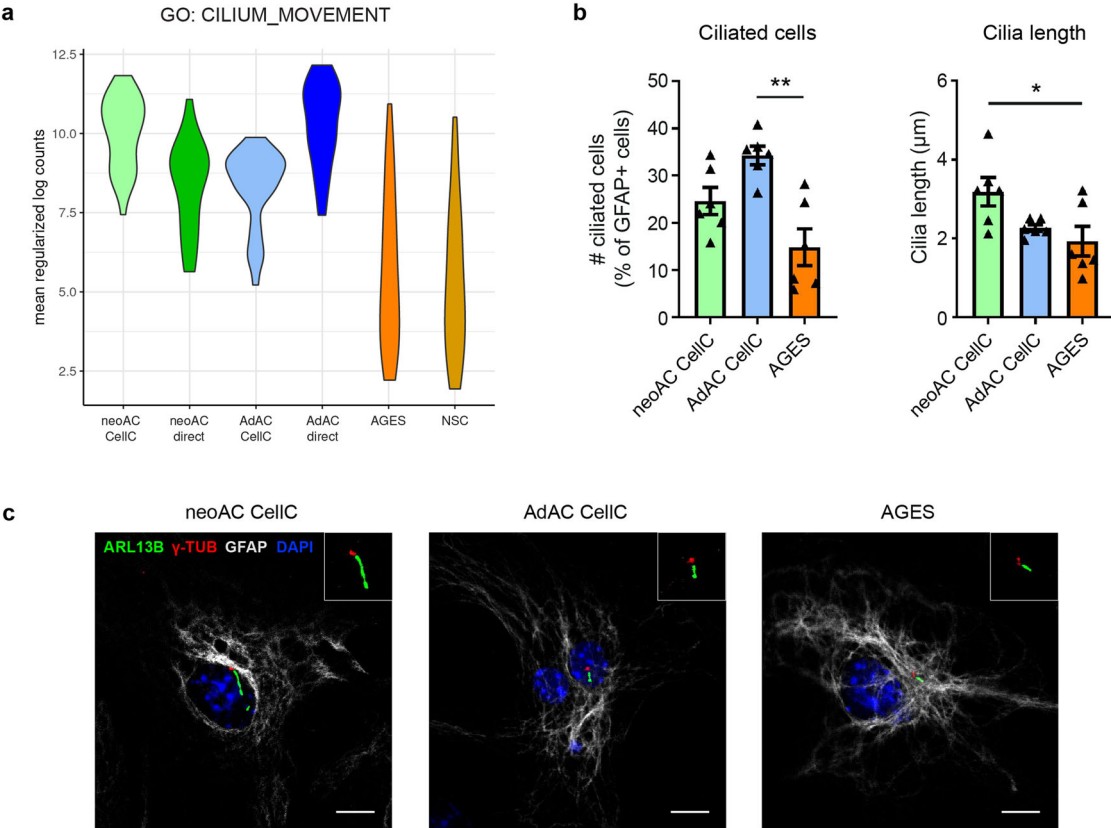

**Fig. 4 Primary cilia morphology and appearance was affected by age and cell origin. a** Comparison of the gene expression for 35 genes in the GO term cilium movement by displaying the mean regularized log counts. neoAC CellC vs. neoAC direct **$P = 0.0025$; AdAC CellC vs. AdAC direct ***$P < 0.001$; neoAC CellC vs. AdAC CellC **$P = 0.0014$; neoAC direct vs. AdAC direct ***$P < 0.001$; AGES vs. neoAC CellC ***$P < 0.001$; AGES vs. AdAC CellC ***$P < 0.001$; AGES vs. NSC non-significant. **b**, **c** Cultured neonatal (neoAC CellC) and adult (AdAC CellC) astrocytes and AGES were immunostained for the cilia marker ARL13B (green) and γ-TUBULIN (red) and counterstained with the astrocyte marker GFAP (white) and the cell nuclei marker DAPI (blue). The percentage of ciliated GFAP-positive cells compared to all GFAP-positive cells, as well as the cilia length in μm, was determined by Fiji analysis. Representative images are shown in (**c**). Scale bar = 10 μm. neoAC CellC ($n = 6$), AdAC CellC ($n = 6$), AGES ($n = 6$); Mean ± SEM, ANOVA with Tukey´s post hoc test; *$P = 0.0264$; **$P = 0.0010$.

AGES could provide an explanation for their differential response to wound gaps, we quantified the number of ciliated cells and the cilia length in cultured neonatal and adult astrocytes as well as in AGES by performing immunostaining for the cilia marker ARL13B. The number of ciliated cells was significantly lower in AGES than those in adult astrocytes, whilst the cilia length of AGES was significantly shorter than in neonatal astrocytes (Fig. 4b, c), validating the observed RNA-seq differences. Whilst the presence of primary cilia has not yet been described in stem cell-derived astrocytes, our results indicate that cilia properties in AGES compared to neonatal and adult primary astrocytes are different, thus potentially accounting for the observed functional differences.

## Conclusion

In summary, our in-depth molecular side-by-side analysis of directly isolated and cultured mouse primary astrocytes and AGES showed clear differences in astrocyte subsets and marker profiles, functional readouts, and transcriptomic signatures. Physiological in vivo functions driven by intercellular interactions are lost in cultured primary astrocytes resulting in distinct transcriptomic profiles. Although in vitro modeling of astrocytes is an inevitable tool which has been crucial for vital advances and discoveries, our data highlight the need to better mimic the CNS microenvironment in vitro when investigating trophic support provided by astrocytes. Mimicking a CNS microenvironment in culture might also assist in differentiating ESCs or iPSCs into

mature astrocytes with closer similarity to primary astrocytes. Whilst AGES seem to mimic some functional characteristics of primary astrocytes, their unilateral astrocyte subset, distinct marker and transcriptomic profile, astrocyte (re)activity response, calcium activity, and cilia profile should be considered when choosing this model. Further, the inherent fact that a low number of oligodendrocytes is co-isolated by MACS should be taken into account when interpreting findings using this model system. In our own study, we cannot rule out that the low levels of contaminating oligodendrocytes affected our RNA-seq results, however, the findings aligned well with validation experiments targeting either astrocyte-specific functions or considering astrocyte morphology or marker expression, thus speaking in favor of their validity. Our investigation is also aimed at supplementing future analyses and comparisons of astrocyte differentiation and culturing protocols also in the context of regional astrocytic heterogeneity. Further, our study highlights the importance of making informed decisions about the biological question under investigation and to match it to a suitable astrocyte isolation and cultivation protocol.

## Methods

**Mice**. For adult astrocyte cultures, C57BL/6 J mice at an age of 100–140 days were used, whilst neonatal cells were taken from p 4–8 C57BL/6 J pups. Female and male mice were mixed for all analyses. Mice were group housed under specific pathogen-free conditions on a 12 h light/dark cycle, and food and water were provided to the mice ad libitum. All animal experiments were performed in accordance with the

national animal protection guidelines approved by the regional offices for health and social services in Berlin (LaGeSo, license numbers T 0276/07 and O298/17).

**Adult astrocyte isolation via MACS.** Adult astrocytes were isolated using magnetic-activated cell sorting (MACS; Miltenyi Biotec) according to the manufacturer's protocol. In brief, mice were anesthetized with Isoflurane, euthanized with $CO_2$, the brain dissected and placed in cold Dulbecco's phosphate-buffered saline (D-PBS) supplemented with 0.91 mM $CaCl_2$, 0.49 mM $MgCl_2$-$6H_2O$, 0.55 mM glucose, and 0.033 mM sodium pyruvate (pH 7.2) (termed D-PBS). For direct astrocyte isolation, the olfactory bulb and cerebellum were removed and the brains of two mice were pooled for tissue dissociation. Tissue dissociation was performed in C-tubes (Miltenyi, 130-096-334) using the Adult Brain Dissociation Kit (Miltenyi, 130-107-677) on program 37C_ABDK_01 of the gentleMACS™ Octo Dissociator with Heaters (Miltenyi Biotec, 130-096-427) allowing for simultaneous enzymatic and mechanical tissue disruption. Red blood cells and debris were removed and astrocytes were magnetically labeled using ACSA-2 microbeads (Miltenyi Biotec, 130-097-678) according to the manufacturer's protocol. The resulting cell suspension was filtered once through a 70 µm pre-separation filter, consecutively passed through two MS columns (Miltenyi Biotec, 130-042-201) placed on an OctoMACS™ manual separator and flushed into Eppendorf tubes in 0.5 % BSA in PBS, pH 7.2 or cell culture medium (see the section on astrocyte culture below). Cell pellets were collected, snap-frozen in liquid nitrogen, and stored at −80 °C until further use.

**Neonatal astrocyte isolation via MACS.** Neonatal astrocytes were isolated by using MACS from p 4–8 C57BL/6 J mice. Cerebral tissue was isolated, meninges removed and brains of two mice pooled in a C-tube for tissue dissociation with the Neural Tissue Dissociation Kit (Miltenyi, 130-092-628) on program 37C_NTDK_1 of the gentleMACS™ Octo Dissociator with Heaters (Miltenyi Biotec, 130-096-427) allowing for simultaneous enzymatic and mechanical tissue disruption. Afterward, the cell suspension was washed with Hanks' Balanced Salt solution without calcium and magnesium (HBSS, Thermo Fisher, 14170138) and astrocytes magnetically labeled with ACSA-2 microbeads (Miltenyi Biotec, 130-097-678), according to manufacturer's protocol. The resulting cell suspension was filtered once through a 70 µm pre-separation filter, consecutively passed through two MS columns (Miltenyi Biotec, 130-042-201) placed on an OctoMACS™ manual separator and flushed into Eppendorf tubes in 0.5 % BSA in PBS, pH 7.2 or cell culture medium (see the section on astrocyte culture below). As the myelin content of the neonatal brain is low[46], no myelin removal was performed. Cell pellets were collected, snap-frozen in liquid nitrogen, and stored at −80 °C until further use.

**Adult and neonatal astrocyte culture.** Cell culture plates (24-well or 96-well as indicated; BD Biosciences) were prepared two days prior to astrocyte isolation. First, wells were coated with 20 µg/ml poly-L-lysine (PLL) (Sigma, 2636-25MG) in PBS overnight at 37 °C, 5% $CO_2$. The next day, wells were washed twice with PBS and subsequently coated with 2 µg/ml laminin in PBS (Sigma, L2020) overnight at 37 °C, 5% $CO_2$. For immunofluorescence, cells were plated onto 13 mm coverslips (VWR) coated with 0.5 mg/ml PLL and 10 µg/ml laminin. For calcium imaging, cells were plated onto glass bottom dishes (MatTek, P35G-1.5-14-C) coated with 0.5 mg/ml PLL and 10 µg/ml laminin. Astrocyte isolation was performed as described above under sterile conditions. ACSA-2 labeled cells were flushed from the MS column with pre-warmed AstroMACS medium (Miltenyi Biotec, 130-117-031) supplemented 50 U/ml penicillin/streptomycin (Sigma, P0781-20ML) and 0.25% L-glutamine (0.5 mM; Thermo Fisher, 25030-024). Neonatal and adult astrocytes were cultured using the same medium composition. Cells were plated at 100,000 cells/24-well or glass dish or 25,000 cells/96-well onto the middle of the well in a droplet of AstroMACS medium and incubated at 37 °C, 5% $CO_2$ for 1–3 h before filling up the medium. The medium was changed every three days and grown for 7–10 days before use.

**Astrocyte differentiation from embryonic stem cells.** Astrocytes were differentiated from embryonic stem cells derived from C57Bl/6 N mice (mESCs; GSC-5003, MTI-GlobalStem) using a previously published protocol[16]. mESCs were first differentiated into neural stem cells (NSCs) by plating mESCs as single cells on flasks coated with 0.1% gelatin (Sigma, G9391-100G) in N2B27 medium (50% DMEM/F12 medium (Thermo Fisher, 11320-033), 50% Neurobasal medium (Thermo Fisher, 21103-049) supplemented with 1% N2 supplement (Thermo Fisher, 17502-048), 2% B27 supplement (Thermo Fisher, 17504-044), 2 mM Glutamax (Thermo Fisher, 35050-038), 55 µM 2-mercaptoethanol (Thermo Fisher, 21985-023), 0.075% Insulin (Sigma, I0278-5ML), 50 mg/ml bovine serum albumin (BSA, Thermo Fisher, 15260-037), and 50 U/ml penicillin/streptomycin (Sigma, P0781-20ML). After differentiating mESCs to NSCs for 14 days, cells were detached with 0.05% trypsin and re-plated on 0.1% gelatin-coated flasks as single cells in N2B27 medium supplemented with 20 ng/ml FGF (Peprotech, 450-33) and 20 ng/ml EGF (Peprotech, AF-315-09). At this stage, the NSC culture consisted of an inhomogeneous culture with bipolar, triangular, and aggregating cells (Supplementary Fig. 3a). For selection and maintenance of bipolar NSCs, cells were split at a density of 80–90% by tapping flasks for removing cell aggregates. Cells were washed with PBS once and detached with 0.05% trypsin for 10–15 s. Here, the

more trypsin-sensitive-bipolar NSCs went into suspension while triangular cells remained attached. After 10–15 s of trypsinization, trypsin was diluted with one part PBS and the cell suspension was filtered through a 70 µm cell strainer into a tube prefilled with two parts PBS. Cells were centrifuged at $900 \times g$ for 3 min and resuspended in N2B27 medium supplemented with 20 ng/ml FGF and 20 ng/ml EGF. For all experiments described here, NSCs with a passage number from 50–60 were used to assure full purity.

For differentiating NSCs into astrocytes generated from ESCs (AGES), plates or dishes were coated with 10 µg/ml Poly-L-Ornithine (PLO, Sigma, P4957) in PBS for at least 2 h, wells were washed twice with PBS and subsequently coated with 2 µg/ml laminin (Sigma, L2020) in PBS overnight at 37 °C, 5% $CO_2$. NSCs were detached as described for cell maintenance and seeded at a density of 100,000 cells into precoated 24 wells in N2B27 medium supplemented with 20 ng/ml BMP4 (Peprotech, 315-27). After 3 days of differentiation in BMP4-supplemented N2B27 medium, AGES were used for described experiments. The medium was changed on day 1 of differentiation and every 3 days afterward.

**Microglia-conditioned medium.** Microglia were isolated from p 1–4 C57BL/6 J mice and prepared as described previously[47]. In brief, brains were dissected, meninges removed, and brains mechanically and enzymatically homogenized with 0.005% trypsin/EDTA. Microglia were cultured in DMEM medium (Invitrogen, 41966-029) supplemented with 10% FBS (PAN-Biotech, P40-37500) and 1% penicillin/streptomycin (Sigma, P0781-20ML) at 37 °C with 5% $CO_2$. After 7 days, microglia proliferation was induced by adding 5 ng/ml GM-CSF (Miltenyi Biotec, 130-095-746) to the medium. Microglia were harvested after 10–13 days in culture by manually shaking flasks for 6 min. After a settling time of 24 h, microglia were cultured in the respective astrocyte medium and either treated with LPS (1 µg/ml, Sigma, L4391-1MG) for 24 h or remained non-treated. The microglia-conditioned medium was centrifuged and the supernatant was added to the astrocyte cultures for 24 h.

**Immunocytochemistry and confocal microscopy.** Cells were fixed for 20 min at room temperature (RT) in freshly prepared 4% paraformaldehyde (PFA) in PBS buffer, pH 7.4. For O4 and GFAP staining, membranes were permeabilized with 0.1% Triton-X100 in PBS for 20 min at RT, and samples were blocked with freshly prepared 3% bovine serum albumin (BSA, Merck) in PBS for 1 h at RT. For staining cilia, cells were stained as previously described[48] by incubating them with 0.1% Triton-X100 and 10% horse serum in PBS for 1 h at RT after fixation. GFAP (Abcam, ab4674, 1:1000), O4 (Miltenyi Biotec, 130-115-810, 1:100), ARL13B (Proteintech, 17711-1-AP, 1:1000), and gamma-Tubulin (Sigma, T6557, 1:800) primary antibodies were diluted in the blocking solution and incubated overnight at 4 °C. Following washing with 0.1% Triton-X100 in PBS for 60 min, Alexa Fluor® 488 Goat Anti-Mouse (Thermo Fisher, A11001), Alexa Fluor® 647 Goat Anti-Rabbit (Thermo Fisher, A21244), Alexa Fluor® 568 Donkey Anti-Mouse (Thermo Fisher, A10037), Alexa Fluor® 488 Goat Anti-Chicken (Thermo Fisher, A11039), and Cy™5 Donkey Anti-Chicken (Jackson ImmunoResearch, 703-175-155), secondary antibodies were diluted 1:500 in PBS and incubated at RT for 1 h. Nuclei were stained with 5 µg/ml 4′,6-diamidino-2-phenylindole (DAPI; Roche, 1023627600) in PBS for 1 min, and coverslips were mounted onto SuperFrost® Plus slides (R. Langenbrink) with fluorescent Aqua-Poly/Mount mounting medium (Polysciences, 18606-20). The cells were imaged using a Leica TCS SP5 confocal laser scanning microscope and an HC PL APO lambda blue 63× oil UV objective or HC PL APO 40.0x, 1.30 or HC PLO APO Ibd.BL 20.0x, 0.70 oil objective controlled by the LAS AF scan software (Leica Microsystem, Germany). Three-dimensional image stacks (1 µm step size) were taken and are shown as maximum projections.

**Image analysis.** Analysis of immunostained neonatal astrocytes, adult astrocytes, and AGES was conducted using Fiji (https://fiji.sc). To determine the percentage of GFAP-positive astrocytes, the total number of DAPI stained nuclei was quantified first: maximum intensity projections of Z-stacks of the DAPI channel were thresholded, and the resulting mask further processed using Binary > Fill Holes and Watershed. Cells were then counted using Analyze > Analyze Particles. An unspecific DAPI signal was excluded by setting a minimum particle size. The number of GFAP-negative cells was counted manually and subtracted from the total number of cells to define the number of GFAP-positive cells. O4-positive cells were manually counted and normalized to the number of DAPI-positive cells to receive the percentage of O4-positive cells.

Based on the maximum intensity projections of merged GFAP, ARL13B, and DAPI channels, cells positive for GFAP and ARL13B were counted manually. The resulting count was then divided by the previously defined number of GFAP-positive cells to determine the percentage of ciliated astrocytes. The mean primary cilia length was determined by tracing ARL13B-positive cilia using the freehand line tool and calculating the length using Analyze > Measure.

**FACS analysis of purity.** For fluorescence-activated cell sorting (FACS) of cultured cells, cells were detached in PBS using a cell scraper. For FACS analysis directly after cell isolation, cells were collected after flushing from MS columns. Cells were pelleted by centrifugation at $300 \times g$, 10 min, 4 °C and stained with the

APC-labeled ACSA-2 antibody (1:10, Miltenyi Biotec, 130-102-315) or the FITC-labeled CD11b antibody (1:200, Biolegend, 101206) in 0.5% BSA in PBS, pH 7.2 for 10 min at 4 °C. Cells were washed in 0.5% BSA in PBS, pH 7.2 and collected by centrifugation followed by FACS analysis. Flow cytometry was performed using a FACSCanto II (BD Biosciences) and analyzed with FlowJo 7.6.5 software. Doublets were excluded by gating before analysing ACSA-2 levels.

**Western blot**. Cell pellets were lysed in RIPA (50 mM Tris-HCl (pH 8), 150 mM NaCl, 1% NP40, 0.5% sodium deoxycholate, 1% SDS, 2 mM EDTA) and protein samples buffer (0.12 M Tris-HCl (pH 6.8), 4% SDS, 20% glycerol, 5% β-mercaptoethanol, bromo phenol blue). Proteins were separated by Tris-Tricine polyacrylamide gel electrophoresis (PAGE) and transferred by wet blotting onto a nitrocellulose membrane. The following primary antibodies were added after blocking with 5% skim milk in Tris-buffered saline with 0.5% Tween20: GFAP (Abcam, ab4674, 1:60000, 30 min at RT), GLT-1 (Cell Signaling, 3838 S, 1:1000, overnight at 4 °C), S100B (Abcam, ab41548, 1:500, overnight at 4 °C), GAPDH (EMD Millipore, MAB374, 1:500, overnight at 4 °C). After secondary antibody incubation, the SuperSignal West Femto Maximum Sensitivity Substrate (Thermo Fisher, 34096) was used for signal detection. The respective intensity of each band was quantified by ImageJ and normalized to the GAPDH intensity. The uncropped western blots are shown in Supplementary Fig. 6.

**RNA isolation and quality control**. Total RNA from frozen cell pellets or astrocyte cultures was isolated using the NucleoSpin miRNA kit (Macherey Nagel, 740971-250). RNA fractions containing small and large RNA were isolated and combined according to the manufacturer's instructions. For quantitative real-time PCR experiments, the rDNase treatment was omitted whilst, for sequencing experiments, rDNase treatment was included. RNA was eluted using pre-warmed RNase-free $H_2O$, snap-frozen in liquid nitrogen, and stored at −80 °C until further use. RNA concentration was measured on a NanoQuant Plate™ using an Infinite® 200 Pro plate reader and the i-control™ Microplate Reader Software (Tecan Life Sciences). For sequencing samples, RNA quality was further analysed by determining the RIN using the RNA ScreenTape (Agilent, 5067-5576) measured on the 4200 TapeStation system (Agilent). Samples with a RIN over 7 and a concentration of at least 100 ng total RNA were used.

**Quantitative real-time PCR**. cDNA was synthesized from ≤1 µg RNA using the QuantiTect Reverse Transcription Kit (Qiagen, 205311), snap-frozen, and stored at −80 °C until further use. Gene expression analyses were performed on 12 ng cDNA per reaction using the TaqMan Fast Advanced Master Mix (ABI, 4364103) in a 384-well plate on a QuantStudio™ 6 Flex Real-Time PCR System (Thermo Fisher, A28139). Each PCR cycle had the following conditions for denaturation, annealing, and lastly, extension: 95 °C for 20 s, 95 °C for 1 s, and 60 °C for 20 s. *Gapdh* was used as an internal control and the delta-delta Ct method was used for quantification. Three technical replicates per condition were performed. The following Taqman primers (Thermo Fisher) were used: *Aldh1l1* (Mm03048957_m1), *beta-III-tubulin* (*Tuj1*) (Mm00727586_s1), *C3* (Mm01232779_m1), *Gapdh* (Mm99999915_g1), *Gfap* (Mm01253033_m1), *Mbp* (Mm01266402_m1), *Mx1* (Mm00487796_m1), *Rbfox3* (Mm01248771_m1), *Serpina3n* (Mm00776439_m1), *Slc1a3* (Mm00600697_m1), and *S100b* (Mm00485897_m1).

**Quantitation of cell proliferation**. Cell proliferation was measured based on the ability of living cells to incorporate EdU with the Click-iT™ EdU Microplate Assay (Invitrogen, 10214). The manufacturer's protocol was followed. Briefly, cells cultured at a density of 50,000 cells per 96 wells were incubated with 10 µM EdU diluted in a respective medium for 72 h. Afterward, the medium was removed and collected for the LDH cytotoxicity assay (see below) and cells were fixed and click-labeled. EdU was detected by using an anti-Oregon Green HRP antibody provided in the kit and incubated with an Amplex UltraRed reaction mixture for detection. Fluorescence was measured on the Infinite® 200 Pro plate reader (Tecan Life Sciences) using an emission wavelength of 530 nm and an integration time of 40 µs. Each measurement was normalized to no EdU control wells to normalize for background fluorescence. Two technical replicates per condition were performed.

**Cytotoxicity**. Cytotoxicity was assessed by measuring lactate dehydrogenase (LDH), an enzyme released upon cell lysis, using the CytoTox 96® Non-Radioactive Cytotoxicity Assay (Promega, G1780). As a positive control for 100% cytotoxicity, cells were lysed by adding 10% Triton-X to the medium and incubated for 30 min at 37 °C. For LDH detection, the manufacturer's protocol was followed. In brief, the cell medium of astrocytes used for the cell proliferation assay (see above) was added to the CytoTox 96® Reagent for 30 min before measuring absorbance at 492 nm (600 nm reference value) on an Infinite® 200 Pro plate reader. All values were presented relative to lysed cell positive control. Two technical replicates per condition were performed.

**Wound sensing assay**. A wound gap was created by scratching a straight line through the confluent monolayer of 100,000 cells/well in a 24-well plate using a sterile 200 µl pipette tip. To guarantee imaging of the same spot, a line was drawn on the outside of the well perpendicular to the scratch. The intersection was used as the area to take all photos. Photos were taken with a Carl Zeiss Axio observer Z1 inverted microscope at time point 0 h and every consecutive 24 h to monitor cellular migration. Quantification was performed using the ImageJ (NIH, Maryland, USA) ROI tool, and the area which was not covered by cells was measured and normalized to the non-treated control.

**Glucose uptake**. Glucose uptake in AGES, neonatal and adult astrocytes plated on 96-well plates (50,000 cells per well) was measured with the Glucose Uptake-Glo™ Assay (Promega, J1341) according to the manufacturer's protocol. In brief, the medium was removed and collected for the Lactate-Glo™ Assay (see below). Cells were washed once with PBS and 1 mM 2-deoxyglucose (2DG) in PBS was added for 10 min to induce glucose uptake. The reaction was stopped using Stop Buffer followed by the addition of a Neutralization Buffer and the 2DG6P Detection Reagent. After 2 h, luminescence was measured on the Infinite® 200 Pro plate reader (Tecan Life Sciences) using no attenuation, 1000 ms integration time, and 0 ms settle time. Each measurement was normalized to reference absorbance at 600 nm at 0 h to account for cell number differences. Wells not containing any cells but included in the measurement protocol were used as a background reference. A standard curve ranging from 2.5 to 20 µM was generated with the provided 2DG6P standard to determine the glucose uptake concentration per well. Two technical replicates per condition were performed.

**Lactate release**. Lactate release was measured on cell medium collected from AGES, neonatal, and adult astrocytes using the Lactate-Glo™ Assay (Promega, J5021). An equal volume of Lactate Detection Reagent was added and the plate was shaken for 60 s. After 1 h, luminescence was measured on the Infinite® 200 Pro plate reader (Tecan Life Sciences) using no attenuation, 1000 ms integration time, and 0 ms settle time. Each measurement was normalized to reference absorbance at 600 nm to account for cell number differences and to cell-free medium as a background reference. A standard curve ranging from 3.125 to 200 µM was generated with the provided lactate standard and used to determine the lactate release concentration. Two technical replicates per condition were performed.

**Synaptosome uptake assay**. For the analysis of synaptosome uptake by primary astrocytes and AGES, pH-sensitive labeled synaptosomes were used. Synaptosomes were isolated as previously described[49]. Unperfused brains of C57BL/6 J mice at 65–75 days were placed in half-frozen PBS and cortices removed. Cortices, including hippocampus were cut into small pieces and homogenized with a Teflon Homogenizer in homogenization buffer (10.9% sucrose, 20 mM HEPES, 0.029% EDTA, protease inhibitor, pH 7.4). Afterwards, the homogenized brain was centrifuged for 10 min at 3000 rpm at 4 °C. The synaptosomes in the supernatant were pelleted for another 15 min at 12,500 × g at 4 °C. The pellet containing the synaptosomes was homogenized in a homogenization buffer using the Teflon homogenizer. Afterward, synaptosomes were separated using a gradient of 0.8 M sucrose layered by 1.2 M sucrose and ultracentrifugation at 20,100 × g for 1 h 10 min at 4 °C. The synaptosome band formed between both sucrose layers was collected. Synaptosomes were pelleted by centrifugation at 12,500 × g for 15 min and resuspended in 0.1 M sodium bicarbonate to label them with a pH-sensitive fluorogenic dye (pHrodo™ Red succinimidyl ester, Thermo Fisher, 10676983, 1:660 dilution) by rotation for 2 h at RT. After washing labeled synaptosomes three times, they were resuspended in PBS containing 5% DMSO. Synaptosomes from one brain were resuspended in 100 µl final volume and were diluted 1:12.5 in culture medium and added to the cells for 24 h before reading fluorescence with an Infinite® 200 Pro plate reader at 560 nm excitation wavelength/585 nm emission wavelength.

**Calcium imaging**. Calcium imaging was performed on cells grown on glass bottom dishes (35 mm dish, 14 mm glass; MatTek, 35G-1.5-14-C) using the Fluo-4, AM calcium indicator (Thermo Fisher, F14201). For reconstitution, a 20% Pluronic® F-127 (Sigma, P2443) in DMSO solution was mixed 1:1 with 10 mM Fluo-4AM in DMSO to yield a 10% Pluronic® F-127/5 mM Fluo-4, AM working solution. On the day of imaging, cells were incubated with 5 µM Fluo-4, AM solution dissolved in sterile HEPES buffer (150 mM NaCl, 5.4 mM KCl, 1.3 mM $CaCl_2$, 0.83 mM $MgSO_4 \times 7\ H_2O$, 10 mM HEPES, 5 mM D(+)-Glucose, pH 7.4) for 30 min at RT. Afterward, the Fluo-4, AM was removed and cells were kept in HEPES buffer at 37 °C, 5% $CO_2$ until and during imaging. The dish was transferred to an Okolab Incubation Chamber. Imaging was conducted at the Advanced Medical Bioimaging Core Facility (AMBIO) at the Charité Universitätsmedizin Berlin using the Nikon Spinning Disc Confocal CSU-X setup with the Nikon Eclipse TI microscope. An ATP solution for inducing calcium release in astrocytes was prepared with a final concentration of 100 µM in HEPES buffer. Using the peristaltic perfusion system (Multi Channel Systems, PPS5), the cells were perfused with HEPES for 2 min, followed by a 30 s perfusion with 100 µM ATP and finally 1 min 30 s in HEPES with a flow rate of 2.5 ml/min. Imaging was started after perfusing cells for 1 min 50 s with HEPES. Intracellular $Ca^{2+}$ changes were detected using the NIS Elements Imaging Software (Nikon, Version 5.10). Analysis was conducted using the ImageJ software with the Time Series Analyzer V3 plugin. The magnitude of $Ca^{2+}$

concentration changes was detected via a temporal analysis of 10 single cells per n as a fluorescence intensity ratio F/F0. F0 was determined at a baseline 10 s window prior to ATP administration. For analysis, the average maximum fluorescence intensity, the time until cells responded to ATP treatment, and the peak width (time between maximum intensity and reaching basal levels) was determined.

**Library preparation and RNA sequencing**. RNA sequencing libraries were prepared using the TruSeq™ Stranded mRNA Library Prep kit (20020594, Illumina) starting from 100 ng of total RNA (RIN ≥7) on an ep*Motion*® 5075 TMX workstation (Eppendorf). Library QC included size distribution check (BioAnalyser) and concentration determination with KAPA Library Quantification Kit (KK4857, Roche). Libraries were equimolar pooled and loaded on Illumina NovaSeq 6000 SP flowcell at 300 pM loading concentration with 1% of PhiX by-mix. Sequencing was performed in paired-end 2 × 100 nt sequencing mode with 8-nt index (i7) read.

**Data analysis**. RNA-seq reads were mapped to the mouse genome (GRCm38/p5) with STAR (version 2.7.3a) using the following parameters: --outFilterType BySJout --outFilterMultimapNmax 20 --alignSJoverhangMin 8 --alignSJDBoverhangMin 1 --outFilterMismatchNmax 999 --outFilterMismatchNoverLmax 0.04 --alignIntronMin 20 --alignIntronMax 1000000 --alignMatesGapMax 1000000[50]. We obtained, on average, 87.7% uniquely mapped reads per sample. Reads were assigned to genes with *featureCounts* (-t exon -g gene_id, version 2.0.1) using Gencode GRCm38/vM12 gene annotation[51]. The differential expression analysis was carried out with DESeq2 (version 1.22.1) using default parameters (Supplementary Data 5). For the principal component analysis (PCA) (in Fig. 3b, c), we used rlog (DESeq2, version 1.22.1) transformed counts[52]. Gene ontology enrichment analysis was done with the CERNO algorithm from R tmod (0.46.2) package:[30] for two-sided comparisons (e.g., AdAC CellC vs direct) genes were sorted by their adjusted (Benjamini–Hochberg) *p* values, and for PCA-based enrichment analysis genes were sorted by their PCA loadings. The enrichment analysis was done using GO gene set collection from MsigDB (r msigdbr 6.2.1). Subcategories BP, MF, and CC were combined for the analysis.

**Statistics and reproducibility**. Data were generated as multiple exploratory analyses to generate hypotheses and biostatistical planning for future confirmatory studies. Data analysis, processing, descriptive, and formal statistical testing were done according to the current customary practice of data handling using Excel 2016, GraphPad PRISM 5.0, and ImageJ. Source data of all figures are provided in Supplementary Data 4. All data generated or analysed during this study are included in this article.

Values are presented as mean ± SEM (standard error of the mean), with each dot representing one biological replicate. Biological replicates are defined by the number of isolations for primary neonatal and adult astrocytes (two mice per isolation) and by the number of independent cultures from different passages for AGES and NSCs. Statistical difference between means was assessed either by the two-tailed t-test for two groups or ANOVA with the indicated post hoc test for more than two groups using the GraphPad Prism software. Outliers were not excluded and statistically significant values are indicated as *$P ≤ 0.05$, **$P ≤ 0.01$, and ***$P ≤ 0.001$.

**Reporting Summary**. Further information on research design is available in the Nature Research Reporting Summary linked to this article.

## Data availability
The primary datasets generated during the study are available in Supplementary Data 1–5 and Supplementary Fig. 6. The RNA-seq dataset generated during and/or analysed during the current study are available in the GEO repository, GSE189745[53].

## Code availability
The R code used for the analysis of the current study is available from the corresponding author on request.

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

## Acknowledgements

This project has received funding from the Deutsche Forschungsgemeinschaft (DFG, German Research Foundation) under Germany's Excellence Strategy—EXC-2049—390688087, as well as HE 3130/6-1 to F.L.H., the German Center for Neurodegenerative Diseases (DZNE) Berlin, and the Innovative Medicines Initiative 2 Joint Undertaking under grant agreement No 115976. This Joint Undertaking receives support from the European Union's Horizon 2020 research and innovation program and EFPIA. S.S. was funded by a PhD fellowship of the NeuroCure Excellence Cluster EXC-2049. We are grateful to Alexander Haake for his excellent technical support. Illustrations were created with BioRender.com (Fig. 1a, b, c, h). The senior author (F.L.H.), as a member of the DZNE Berlin, is permitted to publish pictures made by BioRender.

## Author contributions

K.F. and P.E. performed experiments and analyzed data; A.I. and D.B. performed and supervised bioinformatical analyses; N.S. performed the analysis of cilia and western blots; S.S. isolated synaptosomes; T.B. and S. Sauer performed and supervised RNA library preparation and RNA sequencing; F.L.H. designed and supervised the study; K.F., P.E., and A.I. prepared figures. All authors wrote, revised, and approved the manuscript.

## Funding

## Competing interests

The authors declare no competing interests.
