## [Peer Review File · Communications Biology]

Reviewers' comments:

Reviewer #1 (Remarks to the Author):

In this straightforward manuscript, the authors prepared astrocytes from five different sources: 1) freshly isolated astrocytes from P4-8 neonatal mice, 2) neonatal primary astrocytes after 7-10 culture, 3) freshly isolated astrocytes from adult mice, 4) adult primary astrocytes after 7-10 culture, 5) astrocytes generated from embryonic stem cells (AGES). Functional assay and RNAseq analysis revealed the similarity and differences between astrocytes from these different origins. The results will be of interest to the field when considering the choice of astrocytes origins for the respective biological questions. However, most results in the manuscript are preliminary; major revision is needed before its publication in *Commutations biology*. The purity of isolated astrocytes is a big concern. Confirmation on some of the differentially expressed genes is expected.

Major concerns:

1) The purity of astrocytes isolated from the neonate and adult brain is a major concern. The purification method used in this study, especially for the adult brain, is crude compared to the method used in other similar studies (such as Krawczyk et al, *J Neurosci*, 2022). Indeed, Figure S1 shows that contamination of oligodendrocytes and microglia is heavy. Could this be responsible to the observed difference in the functional assay in Fig 2 and RNAseq analysis in Fig 3?

2) A major group of genes differentially expressed are cilium and axoneme proteins. Does this functional impact cilium growth and axoneme extension at the cellular level? The characterizations along this line are expected. The authors should at least perform immunostaining to verify whether these astrocytes grow cilium or not, and to compare difference of cilium morphology/ciliated rate between astrocytes from different origins.

Minor problems:

There are some mistakes when referring to figures: line 475, the legend for Fig S3F.

Reviewer #2 (Remarks to the Author):

A lack of direct and comprehensive analyses of cultured astrocytes makes it difficult to assess their fidelity in relation to astrocytes *in vivo*. Here, the authors compare mESC-derived astrocytes, primary murine neonatal astrocytes, and adult murine astrocytes using immunocytochemistry, rt-PCR, luminescence-based assays, FACS, and RNA sequencing approaches. The results show significant differences among the astrocyte cultures examined. The authors conclude that there are context-specific merits and disadvantages to each of the astrocyte culture methods. The work provides important information for scientists studying astrocyte function in culture. The story should be accepted for publication after clarification of the following points.

1.) The GFAP images shown in Figure 1D-F do not appear significantly different. The rt-PCR quantitation shows a ~3-fold increase in GFAP relative to neoAC CellC astrocytes. Do the authors have an explanation for this discrepancy? Were the images captured using the same exposure times?

2.) Immuno-panned astrocytes obtained from neonates (P4-5) require a myelin debris removal step (Foo et al., 2011). Why did the authors omit myelin/blood removal from neoAC CellC astrocytes--is this not needed using the magnetic isolation approach?

3.) The increased gene expression levels of GFAP observed in neoAC CellC astrocytes (Figure 1G) could be due to increased reactivity. Have the authors assayed neoAC CellC, AdAC CellC, and AGES for

any markers of reactivity (ALDOC, S100B, C3, Serpina3n, etc.)? This would be valuable.

4.) The graph shown in Figure 2A significant EdU fluorescence in neoAC Cells. The authors should include a positive and negative control for this assay. How do these data compare to highly proliferative cells (e.g. HEK293Ts) and non-proliferative cells (e.g. primary neurons)?

5.) Is it possible that contaminating NSCs are contributing to the calcium response by AGES (Figure 2K). The authors should compare the calcium responses to undifferentiated NSCs. [PLEASE use same color legend in 2K-N]

6.) Table S1 is missing?

7.) Line 214: the authors may wish to cite: doi.org/10.7554/eLife.40202.001 (Fig. 1, Arl13B staining) and doi: 10.7554/eLife.67900 may be relevant to the discussion of astrocyte cilia--since "cilia" gene expression is the biggest difference seen in the present paper experiments, perhaps the authors can stain their cells for Arl13B and compare % ciliated cells and cilia lengths. It seems that panned neonatal rat astrocytes have longer cilia in culture than astrocytes in mouse brain (from those 2 refs.)

Minor comment: Do the authors mean underlying instead of "Underling"?

Reviewer #3 (Remarks to the Author):

The authors of this manuscript describe differences between mouse neonatal and adult astrocytes, in culture and directly isolated from brain, and ESC-derived astrocyte cultures. They analyze this with immunostaining, RNAseq, and several functional assays. The conclusion is that these different astrocytes have all unique phenotypes, which is not very surprising. The manuscript is well written, and the methods section is very clear. The statistics are performed well. The data is important for astrocyte researchers and shows that results from astrocyte cultures should be interpreted with care. The experiments are performed well, but a thorough and substantiated conclusion is missing. Which culture method should other researchers use to study astrocyte functions?

- Line 18: In the text and in some figures: please change "glia cells" into either "glial cells" or "glia."
- Line 34: what is meant with intracellular networks? As this should be inside astrocytes...
- By performing the MACS-isolation with ACSA-2 the authors potentially isolate a subset of astrocytes this might explain also the lower percentage of ACSA-2-positive AGES astrocytes in Figure S1. Please discuss.
- Line 74: the authors mention "protein and transcriptomic profiles". In the manuscript, there are only transcriptomic profiles and a few immunostainings. By reading this line I expect a proteomic analysis as well.
- The genetic background of the mice can have quite an effect on the transcriptomic profiles. The authors use C57Bl6 for the acute isolations and cultures. Please add the genetic background of the ESC-cell line/AGES astrocytes.
- In figure S1 it is not clear to me what the unstained panels show? Or are these the percentage of cells in the gate that is used for ACSA-2 positivity? How is it possible to see ACSA-2 positive cells in the unstained condition? Maybe rename the gate?
- In figure S1 G-J. Fold change on the Y-axis is unclear. Fold change compared to what? I would just mention "normalised data". To use only 1 reference gene in qPCR experiments is not good practice. As the astrocytes have also different metabolic states, this could already change the GAPDH level.
- Line 92. Astrocyte cultures were stained for GFAP, nestin, GLAST and GLT-1. The authors state that all culture astrocytes expressed these markers. That is clearly not true for neoAD and adAC (1D and E).

- Does the presence of GFAP not indicate that all astrocytes are reactive? So how does the described method compare to the classical the Vellis method (line 46-47). Please discuss.
- In figure 1G-I. (see my earlier remark on reference genes. "Fold change on the Y-axis is unclear. Fold change compared to what? I would just use "normalized data". To use only 1 reference gene in qPCR experiments is not good practice. As the astrocytes have also different metabolic states, this could already change the GAPDH level."
- The differences between the cell cultures (especially AGES) could have several causes: genetic background, purity of the culture, differentiation state. Three days of differentiation with BMP4 seems rather short to get fully mature astrocytes. A main difference is also the lack of other cells during differentiation in AGES. This might be the explanation for a high GFAP level, and a low Aldh1l1 and GLAST levels.
- AGES are the odd one out (see transcription profile, scratch assay, calcium imaging). What is the evidence that AGES are more suitable in representing in vivo astrocyte function (line 135). In line 182 the authors state that AGES are not yet fully differentiated. Is 3 days with BMP4 too short?
- Line 149: there is no table S1. Is this figure S4?
- Line 198: here the authors hint that the changes in vitro might be mimicking ageing. What is the evidence?
- Line 209: I guess the word adult is missing: "neonatal and adult astrocyte cultures"
- Are the differences caused by differences in extracellular matrix or differences in cell-cell contacts? Please discuss.
- As the authors stress that the classical method of astrocyte cultures induces a reactive phenotype, it would be important to describe whether the presence or absence of a reactive state in the described neonatal and adult astrocyte cultures.
- AGES have the lowest EdU incorporation and migrate similar to in vivo, still, the authors conclude in line 181 that AGES are not fully differentiated into astrocytes. This seems a contradiction.

- Line 241: ad libidum should be ad libitum.
- Line 247: what is D-PBS? What is the D standing for?
- Line 249: two mice were pooled for astrocyte isolations. What does the N then represent in the experiments? And in line 482 the authors state that the biological replicates are defined by the number of used mice. So is a biological replicate of 3 then cells from 6 mice?
- Line 290: add mouse strain of the mESC cells.
- Figure 3F is extremely difficult to understand, please explain it better.

Point-to-Point Response

→ We'd like to thank all reviewers for their thorough evaluation of our manuscript, their appreciation of our work and their valuable feedback. In the following, we refer to all the reviewers' comments and present additional experiments we performed based on their suggestions. Accordingly, we adjusted parts of the manuscripts and changes were depicted in blue font color in the revised version of the manuscript. We performed a major revision by adding an in-depth purity analysis of all analyzed cell types and a quantification of astrocyte markers on protein level to complement the previous mRNA-based analysis. We also added an analysis of the reactivity state of presented astrocyte cell types as well as a cilia morphology quantification to validate our RNA-seq findings. Additionally, several control experiments were performed to support the validity of the respective experiments. We hope that the revised version of the manuscript now meets the reviewers' and the editors' expectations and presents the missing details and matureness.

Reviewer: 1

"In this straightforward manuscript, the authors prepared astrocytes from five different sources: 1) freshly isolated astrocytes from P4-8 neonatal mice, 2) neonatal primary astrocytes after 7-10 culture, 3) freshly isolated astrocytes from adult mice, 4) adult primary astrocytes after 7-10 culture, 5) astrocytes generated from embryonic stem cells (AGES). Functional assay and RNAseq analysis revealed the similarity and differences between astrocytes from these different origins. The results will be of interest to the field when considering the choice of astrocytes origins for the respective biological questions. However, most results in the manuscript are preliminary; major revision is needed before its publication in *Commutations biology*. The purity of isolated astrocytes is a big concern. Confirmation on some of the differentially expressed genes is expected."

→ We thank reviewer 1 for raising these critical points. Based on reviewer 1's suggestions, we revised the manuscript substantially by performing additional cell purity analyses and by validating our RNA-seq findings to provide mechanistic insights into functional changes observed between the various astrocyte cell types.

Major concerns:

1) The purity of astrocytes isolated from the neonate and adult brain is a major concern. The purification method used in this study, especially for the adult brain, is crude compared to the method used in other similar studies (such as Krawczyk et al, *J Neurosci*, 2022). Indeed, Figure S1 shows that contamination of oligodendrocytes and microglia is heavy. Could this be responsible to the observed difference in the functional assay in Fig 2 and RNAseq analysis in Fig 3?

→ As reviewer 1 rightly stated, our purity analysis of MACS-isolated neonatal and adult astrocytes revealed a certain level of impurity on mRNA level. To determine the level of impurity on cellular level, we determined the number of CD11b-positive microglia in all our isolations by FACS. In comparison to the RT-qPCR data, our FACS analysis revealed that no microglia are present in any of our isolations (new Fig. S1D). As studies also found *Sall1* expression in progenitor cells (Harrison, 2012, *Dis Model Mech.*), we consider *Sall1* not an ideal marker for analyzing the extent of microglia contamination. Therefore, we replaced the previous RT-qPCR analysis by the FACS analysis. To further extend these analyses aimed at defining the purity of astrocytes, the percentage of GFAP-positive and O4-positive cells was determined by fluorescent staining of cultured adult and neonatal astrocytes as well as AGES. Contrary to the finding that AGES contained the lowest number of ACSA-2 positive cells (72 %) as shown by FACS (Fig. S1B), AGES have the highest purity of 98 % GFAP-positive cells. Cultured neonatal and adult astrocytes contain around 79 % and 74 % GFAP-positive cells (new Fig. S1C).

Immunostaining revealed the presence of 1 – 5 % O4-positive oligodendrocytes in MACS-isolated primary astrocyte cultures with no O4-positive oligodendrocytes in the AGES culture (new Fig. S1H), which is in line with the ACSA-2 purity analysis and indicates the presence of GFAP-negative astrocytes in the cultures. Therefore, the *Mbp* RT-qPCR analysis can only be interpreted in terms of relative differences between all isolations. Importantly, as the aim of this study was not to develop a novel astrocyte isolation protocol with higher purity levels compared to previously established protocols, but to assess whether MACS-isolated astrocytes qualify to be regarded as a trustful and authentic model system, we consider it of utmost importance to report the inherent fact that some oligodendrocytes are present in those cultures. These results are supposed to raise awareness in the community using this protocol to adapt their experimental design according to the given scientific question.

However, we regard the influence of these few numbers of contaminating oligodendrocytes on our functional analyses as rather low as these were based on astrocyte-specific behaviors such as cell migration or calcium signaling, where only cells with astrocyte morphology were analyzed. However, we formally cannot exclude that the sole presence of the very few oligodendrocytes may affect astrocytic behavior or RNA-seq profile of these cell types. To implement these findings in our validation experiments, we particularly analyzed the number of ciliated cells and cilia in GFAP-positive cells (new Fig. 4B-C). As we regard this topic of high relevance, we included it in the revised version of our manuscript (line 271-275).

2) A major group of genes differentially expressed are cilium and axoneme proteins. Does this functional impact cilium growth and axoneme extension at the cellular level? The characterizations along this line are expected. The authors should at least perform immunostaining to verify whether these astrocytes grow cilium or not, and to compare difference of cilium morphology/ciliated rate between astrocytes from different origins.

→ Based on these suggestions, we quantified the number of ciliated cells and the cilia length in cultured neonatal and adult astrocytes as well as in AGES by performing immunostaining for the cilia marker ARL13B. This quantification showed that the number of ciliated AGES is lower than the number of ciliated cells in adult astrocytes and the cilia length is significantly shorter than in neonatal astrocytes (new Fig. 4B-C), validating the observed differences seen by RNA-seq. To give further insights into the different regulation of cilia-related genes in all cell types, the regulated log counts of 35 genes of the gene set “GO: cilium movement” were plotted. Interestingly, major differences between AGES and primary astrocytes become apparent. The violin plot of AGES has high similarities with the one of the precursors NSCs, indicating that AGES did not reach the full maturity level with regard to cilia formation (new Fig. 4A). While significant differences were also found between neonatal and adult astrocytes by RNA-seq, no significant difference between neonatal and adult astrocytes was observed by staining for cilia. These data were added to the manuscript as a new Fig. 4 (line 240-257).

Minor problems:

There are some mistakes when referring to figures: line 475, the legend for Fig S3F.

→ The mistake was corrected.

Reviewer: 2

A lack of direct and comprehensive analyses of cultured astrocytes makes it difficult to assess their fidelity in relation to astrocytes in vivo. Here, the authors compare mESC-derived astrocytes, primary murine neonatal astrocytes, and adult murine astrocytes using immunocytochemistry, rt-PCR, luminescence-based assays, FACS, and RNA sequencing approaches. The results show significant differences among the astrocyte cultures examined. The authors conclude that there are context-

specific merits and disadvantages to each of the astrocyte culture methods. The work provides important information for scientists studying astrocyte function in culture. The story should be accepted for publication after clarification of the following points.

→ We thank reviewer 2 for the positive feedback and for his/her thorough review of our manuscript. Based on the mentioned suggestions, we performed additional experiments to address the points raised by him/her.

1.) The GFAP images shown in Figure 1D-F do not appear significantly different. The rt-PCR quantitation shows a ~3-fold increase in GFAP relative to neoAC CellC astrocytes. Do the authors have an explanation for this discrepancy? Were the images captured using the same exposure times?

→ All astrocytic markers were only quantified by RT-qPCR on mRNA level or by FACS on protein level. The images in Fig. 1D-F of the initially submitted manuscript served the pure purpose to visualize glial morphology and to illustrate that these markers are present in all astrocytes. Based on this critique, we changed the figure accordingly: to visualize the morphology, GFAP-stained astrocytes are shown in Fig. 1A-C. Additionally, we now quantified the levels of GFAP, S100b and GLT-1 by Western Blot on protein level, also aimed at validating the mRNA data. These additional quantifications revealed that AGES indeed express higher levels of GFAP than all other cell types (new Fig. 1D), while directly isolated adult astrocytes have higher levels of S100b (new Fig. 1E). GLT-1 protein levels were comparable between all cell types (new Fig. 1F).

2.) Immuno-panned astrocytes obtained from neonates (P4-5) require a myelin debris removal step (Foo et al., 2011). Why did the authors omit myelin/blood removal from neoAC CellC astrocytes--is this not needed using the magnetic isolation approach?

→ Based on current literature, the myelination of mouse brains starts at postnatal days P7 and reaches its peak around P14 (Nishiyama et al. 2021, Seminars in Cell & Developmental Biology). With increasing age, the myelin content as well as the circulatory system including blood vessels increases, making a distinct dissociation and isolation of astrocytes necessary. For this reason, Miltenyi Biotec recommends using the neural tissue brain dissociation kit without myelin debris removal step for mice younger than P7 and the adult brain dissociation kit with myelin debris and red blood cell removal for mice older than P7. We therefore followed the respective manufacturer's recommendations in order to allow comprehensive comparisons using standardized protocols known in the field.

3.) The increased gene expression levels of GFAP observed in neoAC CellC astrocytes (Figure 1G) could be due to increased reactivity. Have the authors assayed neoAC CellC, AdAC CellC, and AGES for any markers of reactivity (ALDOC, S100B, C3, *Serpina3n*, etc.)? This would be valuable.

→ We now included the assessment of the suggested reactivity markers C3, *Serpina3n* and *Mx1*. Astrocytes isolated and cultured in FCS-containing medium based on the McCarthy & de Vellis method (McCarthy, de Vellis, 1980, J Cell Biol.) were included as a positive control. While the McCarthy & de Vellis neonatal astrocytes exhibit a high expression of *Mx1*, *Serpina3n* and *C3*, the astrocyte cell types investigated here presented a very low expression of these (re)activity markers. Interestingly, AGES expressed much higher levels of *C3* compared to all primary astrocytes (new Fig. 1G). The complement factor C3 is known to drive the differentiation of the NSCs (Shinjyo et al. 2009, Stem Cells), thus indicating that the *C3* expression in AGES is still a remnant of the differentiation process. As *Mx1* and *Serpina3n* are lowly expressed, we conclude that the (re)activity state of the AGES is nonetheless rather low. To verify our conclusion we induced (re)activity in all cultured astrocytes using microglia-conditioned medium (MCM), for which microglia were either treated with LPS or remained untreated for 24h. The medium containing all microglial cytokines released upon LPS exposure was then added for 24h to the respective astrocyte culture. Compared to MCM-untreated cells, all astrocyte cultures including neonatal and adult astrocytes as well as AGES massively upregulated *Mx1*, *Serpina3n* and *C3* expression, highlighting that all cultured astrocytes have

a very low baseline (re)activity state. Interestingly, the increase in *Mx1* expression was around 20-fold higher in AGES than in neonatal and adult astrocytes, while the upregulation of *C3* was highest in neonatal astrocytes (new Fig. 1H).

4.) The graph shown in Figure 2A significant EdU fluorescence in neoAC CellCs. The authors should include a positive and negative control for this assay. How do these data compare to highly proliferative cells (e.g. HEK293Ts) and non-proliferative cells (e.g. primary neurons)?

→ Following the suggestions of the reviewer, we added Hek293 cells as a positive control and compared the proliferation rate of Hek293 cells and AGES revealing a massive 100-fold difference in their proliferation rate. AGES are terminally differentiated cells, which show almost no proliferation/fluorescent EdU signal in comparison to the background fluorescent signal measurements for which no EdU was added to the cells. Compared to Hek293 cells, the proliferation rate of neoAC CellC is small but significantly higher than in the non-proliferating AGES (Fig. 2A). We added this control experiment to Fig. S3B.

5.) Is it possible that contaminating NSCs are contributing to the calcium response by AGES (Figure 2K). The authors should compare the calcium responses to undifferentiated NSCs. [PLEASE use same color legend in 2K-N]

→ To assess the purity of our AGES cultures, the percentage of GFAP positive cells of all DAPI-stained cells was determined by fluorescent staining. On average 98 % of all DAPI-positive cells are GFAP-positive, indicating a very high differentiation rate with vanishingly small NSC contaminations (new Fig. S1C). Nonetheless, we compared the calcium response of AGES and NSCs and found a lower maximum fluorescent signal in NSCs than in AGES, underlining that minor NSC contamination would rather decrease than increase the overall calcium response of AGES (new Fig. S3D).

6.) Table S1 is missing?

→ Table S1 was now submitted with the revised manuscript.

7.) Line 214: the authors may wish to cite: doi.org/10.7554/eLife.40202.001 (Fig. 1, Arl13B staining) and [doi: 10.7554/eLife.67900](https://doi.org/10.7554/eLife.67900) may be relevant to the discussion of astrocyte cilia--since "cilia" gene expression is the biggest difference seen in the present paper experiments, perhaps the authors can stain their cells for Arl13B and compare % ciliated cells and cilia lengths. It seems that panned neonatal rat astrocytes have longer cilia in culture than astrocytes in mouse brain (from those 2 refs.)

→ We thank reviewer 2 for the valuable protocol for staining cilia. Based on his/her suggestions, we quantified the number of ciliated cells and the cilia length in cultured neonatal and adult astrocytes as well as in AGES by performing immunostaining for the cilia marker ARL13B. This quantification showed that the number of ciliated AGES is lower than in adult astrocytes and the cilia length is significantly shorter than in neonatal astrocytes validating the observed differences seen by RNA-seq. Interestingly, no significant difference was observed between neonatal and adult astrocytes (see also reviewer 1, point 2). These data were added to the manuscript as a new Fig. 4B-C (line 240-257).

Minor comment: Do the authors mean underlying instead of "Underling"?

→ This typo was corrected.

Reviewer: 3

The authors of this manuscript describe differences between mouse neonatal and adult astrocytes, in culture and directly isolated from brain, and ESC-derived astrocyte cultures. They analyze this with

immunostaining, RNAseq, and several functional assays. The conclusion is that these different astrocytes have all unique phenotypes, which is not very surprising. The manuscript is well written, and the methods section is very clear. The statistics are performed well. The data is important for astrocyte researchers and shows that results from astrocyte cultures should be interpreted with care. The experiments are performed well, but a thorough and substantiated conclusion is missing. Which culture method should other researchers use to study astrocyte functions?

→ We appreciate the thorough and precise reviewing of our manuscript and the positive feedback of reviewer 3. Based on additional experiments showing a clear difference in the number and morphology of cilia in AGES compared to primary astrocytes, we now provide a more substantiated conclusion and include a suggestion, which culture methods may be suited best for assessing certain astrocyte functions. As the use of the model system highly depends on the biological question, the overview in Fig. S5 is supposed to help with choosing the most suitable model system for the respective functional, morphological or marker profile assessment.

- Line 18: In the text and in some figures: please change “glia cells” into either “glial cells” or “glia.

- Line 209: I guess the word adult is missing: “neonatal and adult astrocyte cultures”

- Line 241: ad libidum should be ad libitum.

- Line 74: the authors mention “protein and transcriptomic profiles”. In the manuscript, there are only transcriptomic profiles and a few immunostainings. By reading this line I expect a proteomic analysis as well.

→ We thank reviewer 3 for listing all typos. All of them were corrected in the revised version of the manuscript. We included an additional assessment of the astrocytic marker profile on protein level by Western Blot (new Fig. 1D-F). The wording in line 74 was rephrased to avoid the induction of the expectation that proteomic analyses were performed as part of our analyses.

- Line 34: what is meant with intracellular networks? As this should be inside astrocytes...

→ We intended to imply that astrocytes build a local signaling hub by not only forming physical interactions but also by molecular exchanges with neighboring cells. We agree that this was not clearly specified and we therefore rephrased the text accordingly.

-By performing the MACS-isolation with ACSA-2 the authors potentially isolate a subset of astrocytes this might explain also the lower percentage of ACSA-2-positive AGES astrocytes in Figure S1. Please discuss.

→ We agree with the reasoning of reviewer 3. In addition to the percentage of ACSA-2-positive cells, the percentage of GFAP-positive AGES was determined by staining. The number of GFAP-positive cells as a percentage of DAPI-positive stained cells was on average around 98 % (Fig. S1C), revealing a highly pure AGES culture. As reviewer 3 rightly predicted, this analysis shows that the percentage of GFAP-positive cells does not full overlap with the percentage of ACSA-2-positive cells. This valid and important point is now addressed in the discussion of the revised version of our manuscript (line 96).

- The genetic background of the mice can have quite an effect on the transcriptomic profiles. The authors use C57Bl6 for the acute isolations and cultures. Please add the genetic background of the ESC-cell line/AGES astrocytes.

- Line 290: add mouse strain of the mESC cells.

→ mESCs were derived from C57Bl/6N mice (#GSC-5003, MTI-GlobalStem). This information was added to the method section.

- In figure S1 it is not clear to me what the unstained panels show? Or are these the percentage of cells in the gate that is used for ACSA-2 positivity? How is it possible to see ACSA-2 positive cells in the unstained condition? Maybe rename the gate?

→ We restructured Fig. S1 by only showing the percentage of ACSA-2-positive cells of all cell types next to each other. The gating strategy was moved to Fig. S2. Fig. S2 starts with the SSC vs. FSC to define the cell population of interest (1st dot plot of each row). Afterwards doublets were excluded (2nd dot plot). Cells not labelled with ACSA-2-APC were referred to as “unstained” control (3rd dot plot). Using the unstained control, the gate for ACSA-2 positive cells was defined. All unstained controls contained less than <1 % cells. To determine the percentage of ACSA-2-positive cells (Fig. S1D), the same gating strategy was used for all cell types. To clarify our procedure, we adjusted the figure legend.

- In figure S1 G-J. Fold change on the Y-axis is unclear. Fold change compared to what? I would just mention “normalised data”. To use only 1 reference gene in qPCR experiments is not good practice. As the astrocytes have also different metabolic states, this could already change the GAPDH level.

- In figure 1G-I. (see my earlier remark on reference genes. “Fold change on the Y-axis is unclear. Fold change compared to what? I would just use “normalized data”. To use only 1 reference gene in qPCR experiments is not good practice. As the astrocytes have also different metabolic states, this could already change the GAPDH level.”

→ The RT-qPCR data were normalized to *Gapdh* and to neoAC CellC in Fig. S1I-L as well as to the whole brain samples in Fig. S1E-F. To minimize the text length on the y-axis, the entire normalization procedure is explained in each figure legend. During establishment procedures, *Gapdh* was defined as a reliable housekeeping gene in all astrocyte samples. As a control, we performed the RT-qPCR for the reactivity markers *C3* and *Serpina3n* using two housekeeping genes: *Gapdh* and *Actin*. All relative differences remained very similar (Fig. R1). Thus, we concluded that *Gapdh* is a valid housekeeping gene for our samples.

Figure R1. Housekeeping gene control.

Gene expression of the markers (A) *C3* and (B) *Serpina3n* was determined by quantitative real-time PCR. All expression values were normalized to the internal control *Gapdh* and *Actin* and the fold change compared to cultured neonatal astrocytes (neoAC CellC) was displayed. Mean \pm SEM.

- Line 92. Astrocyte cultures were stained for GFAP, nestin, GLAST and GLT-1. The authors state that all culture astrocytes expressed these markers. That is clearly not true for neoAD and adAC (1D and E).

→ See answer to point 1 of reviewer 2.

- Does the presence of GFAP not indicate that all astrocytes are reactive? So how does the described method compare to the classical the Vellis method (line 46-47). Please discuss.

→ We thank reviewer 3 for raising this interesting point. Even though GFAP upregulation has been found in many CNS disease settings and has been associated with astrocyte reactivity for years, it is nowadays known that GFAP levels also vary in the healthy (mouse) brain. Region-specific basal GFAP expression was described showing higher GFAP content in hippocampal astrocytes (Griemsmann et al. 2015, Cereb. Cortex). Furthermore, GFAP expression fluctuates during development (Riol et al. 1992, J Neurosci Res) and upon environmental changes (Rodríguez et al. 2013, Cell Death Dis). In mice, *Gfap* mRNA expression starts at E14 and reaches its peak at P3. Afterwards, *Gfap* mRNA expression significantly decreases in adult mice before increasing in mice older than 200 days (Riol et al. 1992, J Neurosci Res). As these data nicely correlate with our observations that *Gfap* expression is higher in neonatal astrocytes than adult astrocytes (Fig. S1I), we assume that the variance in *Gfap* expression is due to differences in developmental stages as well as microenvironmental differences. Compared to astrocytes isolated with the classical McCarthy & de Vellis method, all astrocytes reveal a very

low (re)activity state, thus indicating that the differences in *Gfap* do not reflect astrocytic (re)activity (see answer to point 3 of reviewer 2 and new Fig. 1G-H).

- The differences between the cell cultures (especially AGES) could have several causes: genetic background, purity of the culture, differentiation state. Three days of differentiation with BMP4 seems rather short to get fully mature astrocytes. A main difference is also the lack of other cells during differentiation in AGES. This might be the explanation for a high GFAP level, and a low *Aldh1l1* and *GLAST* levels.

- AGES are the odd one out (see transcription profile, scratch assay, calcium imaging). What is the evidence that AGES are more suitable in representing *in vivo* astrocyte function (line 135). In line 182 the authors state that AGES are not yet fully differentiated. Is 3 days with BMP4 too short?

→ We agree with reviewer 3 that the lack of the microenvironment created by other brain cells is most likely the reason for AGES to be “the odd one out” in terms of astrocytic marker profile, transcriptional profile and calcium imaging. In line 135 we particularly refer to the migration behavior of cultured astrocytes: “Despite showing differences to primary astrocytes, AGES might therefore be more suitable in modelling the *in vivo*-like migration behavior of astrocytes.” While the differences in migration behavior of AGES might be due to the lack of a physiological brain microenvironment, they might still represent an attractive model for responses to scratch wounds. Interestingly, cell migration is rather a phenomenon occurring in astrocyte cultures *in vitro*, since *in vivo* rather an extension of processes towards the lesion or a selective proliferation of a subset of astrocytes instead of an active migration has been reported (Bardehle et al. 2013, Nature Neurosci). Even though AGES are not representing the migration behavior of the primary astrocytes, they might still be a good model for representing the extension of processes towards a lesion without active migration reported *in vivo*. We extended the explanation of this phenomenon in the manuscript.

To assess whether the time period for the differentiation of AGES has an effect on the expression of astrocytic markers, we previously performed a time line investigating *Aldh1l1* and *Nestin* expression after 3, 5, 7 and 9 days of differentiation. No significant differences in the expression of *Aldh1l1* and *Nestin* were observed over time (Fig. R2). A tendency of a slightly higher expression of *Aldh1l1* was found at 5 days, which was the differentiation time used for most of the experiments. We therefore conclude that the lack of matureness of AGES is not due to the short time of differentiation.

Figure R2. Astrocytic marker expression changes upon prolonged differentiation of AGES. Gene expression of the astrocytic markers (A) *Aldh1l1* and (B) *Nestin* was determined by quantitative real-time PCR. All expression values were normalized to the internal control *Gapdh* and the fold change compared to AGES day 3 was displayed. Mean \pm SEM, ANOVA with Tukey's post hoc test, $P > 0.05$.

- Line 149: there is no table S1. Is this figure S4?

→ Table S1 was now submitted with the revised version of the manuscript. It is distinct from figure S4 and contains the genes used for the astrocyte marker gene PCA plot in Fig. S4A.

- Line 198: here the authors hint that the changes *in vitro* might be mimicking ageing. What is the evidence?

→ “In summary, the changes introduced *in vitro* thus reflect the change in microenvironment upon culturing in an age-dependent manner since the metabolic support of other cells is no longer required.” With this sentence, we intended to underline that both neonatal and adult astrocytes express genuine differences upon culturing. Interestingly, both neonatal and adult astrocytes respond differently indicating that an age-specific difference is kept even after several days *in vitro*. To clarify our statement, the sentence was rephrased.

-Are the differences caused by differences in extracellular matrix or differences in cell-cell contacts? Please discuss.

→ To address these questions, we looked for gene sets associated with extracellular matrix or cell-cell contacts in the gene set enrichment analysis of the performed RNA-seq. Indeed, the GO term “Regulation of extracellular matrix assembly” revealed differences for the comparison of “AGES vs AdAC CellC” as well as “neoAC CellC vs AdAC CellC”. Given that astrocyte-produced extracellular matrix was described to regulate astrocyte proliferation and migration (Wiese, 2012, Front Pharmacol.), the differences in migration behavior between “AGES vs AdAC CellC” as well as “neoAC CellC vs AdAC CellC” might also be explained by differential production of extracellular proteins. We added this GO term to Fig. 3F. No gene set enrichment terms related to cell-cell contacts were found to be differentially regulated, however, differences in those could still potentially account for the differences in cell migration behavior.

- As the authors stress that the classical method of astrocyte cultures induces a reactive phenotype, it would be important to describe whether the presence or absence of a reactive state in the described neonatal and adult astrocyte cultures.

→ See answer to point 3 of reviewer 2.

- AGES have the lowest EdU incorporation and migrate similar to *in vivo*, still, the authors conclude in line 181 that AGES are not fully differentiated into astrocytes. This seems a contradiction.

→ Based on the extensive explanation given above, we think that these two statements are not contradictory. Indeed, the migration behavior of AGES seems to be similar to the migration behavior described for astrocytes *in vivo* (Bardehle et al. 2013, Nature Neurosci). Based on our RNAseq data, which show a clear separation of primary astrocytes and AGES, we conclude that AGES did not reach a full maturation stage which is most likely due to the lack of the physiological brain microenvironment. Despite their differences, they might still represent a suitable model for assessing the *in vivo*-like non-migratory response to lesions in culture (line 153-156).

- Line 247: what is D-PBS? What is the D standing for?

→ It stands for Dulbecco's phosphate-buffered saline. The explanation of this abbreviation was added to the method section.

- Line 249: two mice were pooled for astrocyte isolations. What does the N then represent in the experiments? And in line 482 the authors state that the biological replicates are defined by the number of used mice. So is a biological replicate of 3 then cells from 6 mice?

→ As correctly stated by the reviewer, the N represents the number of isolations meaning for 6 mice the biological replicates would be 3 as two mouse brains were pooled for 1 isolation. We corrected the statistics section accordingly.

- Figure 3F is extremely difficult to understand, please explain it better.

→ We added some introductory sentences to facilitate the understanding of Fig. 3F (line 184-188). We hope that these improved the understanding of the respective section.

Reviewer #1 (Remarks to the Author):

The authors addressed most of my questions, and added a new figure on the characterization of primary cilia in the astrocytes from different sources. The revised version of the manuscript is significantly improved. I have no further questions and will recommend publication.

Reviewer #2 (Remarks to the Author):

The authors have addressed the concerns of reviewer 2. Thanks!